# Genetic perturbation of PU.1 binding and chromatin looping at neutrophil enhancers associates with autoimmune disease

Stephen Watt[1], Louella Vasquez[1,17], Klaudia Walter [1,17], Alice L. Mann[1,17], Kousik Kundu [1,2], Lu Chen[1,2,13], Ying Sims[1], Simone Ecker [3], Frances Burden[4,5], Samantha Farrow[4,5], Ben Farr[1], Valentina Iotchkova [1,6,14], Heather Elding[1], Daniel Mead [1], Manuel Tardaguila[1], Hannes Ponstingl [1], David Richardson [6], Avik Datta[6], Paul Flicek [6], Laura Clarke [6], Kate Downes[4,5], Tomi Pastinen[7], Peter Fraser [8,15], Mattia Frontini[4,5,9,16,18], Biola-Maria Javierre [8,10,18✉], Mikhail Spivakov [8,11,12,18✉] & Nicole Soranzo [1,2,18✉]

Neutrophils play fundamental roles in innate immune response, shape adaptive immunity, and are a potentially causal cell type underpinning genetic associations with immune system traits and diseases. Here, we profile the binding of myeloid master regulator PU.1 in primary neutrophils across nearly a hundred volunteers. We show that variants associated with differential PU.1 binding underlie genetically-driven differences in cell count and susceptibility to autoimmune and inflammatory diseases. We integrate these results with other multi-individual genomic readouts, revealing coordinated effects of PU.1 binding variants on the local chromatin state, enhancer-promoter contacts and downstream gene expression, and providing a functional interpretation for 27 genes underlying immune traits. Collectively, these results demonstrate the functional role of PU.1 and its target enhancers in neutrophil transcriptional control and immune disease susceptibility.

[1] Human Genetics, Wellcome Sanger Institute, Genome Campus, Hinxton, UK. [2] School of Clinical Medicine, University of Cambridge, Cambridge, UK. [3] UCL Cancer Institute, London, UK. [4] Department of Haematology, University of Cambridge, Cambridge, UK. [5] National Health Service Blood and Transplant (NHSBT), Cambridge, UK. [6] European Molecular Biology Laboratory, European Bioinformatics Institute, Hinxton, Cambridge, UK. [7] Center for Pediatric Genomic Medicine, Children's Mercy, Kansas City, MO, USA. [8] Nuclear Dynamics Programme, Babraham Institute, Cambridge, UK. [9] British Heart Foundation Centre of Excellence, Division of Cardiovascular Medicine, Addenbrooke's Hospital, Cambridge, UK. [10] Josep Carreras Leukaemia Research Institute, Badalona, Barcelona, Spain. [11] Functional Gene Control Group, MRC London Institute of Medical Sciences (LMS), London, UK. [12] Institute of Clinical Sciences, Imperial College Faculty of Medicine, London, UK. [13] Present address: Key Laboratory of Birth Defects and Related Diseases of Women and Children, Department of Laboratory Medicine, West China Second University Hospital, State Key Laboratory of Biotherapy, Sichuan University, Chengdu, China. [14] Present address: MRC Weatherall Institute of Molecular Medicine, Radcliffe Department of Medicine, University of Oxford, John Radcliffe Hospital, Headington, Oxford, UK. [15] Present address: Department of Biological Science, Florida State University, Tallahassee, FL, USA. [16] Present address: Institute of Biomedical & Clinical Science, College of Medicine and Health, University of Exeter Medical School, RILD Building, Exeter, UK. [17] These authors contributed equally: Louella Vasquez, Klaudia Walter, Alice L. Mann. [18] These authors jointly supervised this work: Mattia Frontini, Biola-Maria Javierre, Mikhail Spivakov, Nicole Soranzo. ✉email: bmjavierre@carrerasresearch.org; mikhail.spivakov@lms.mrc.ac.uk; ns6@sanger.ac.uk

Neutrophils are the most abundant circulating leucocytes in human blood, comprising up to 80% of the white blood cell population. They form the body's first line of defence against infection from pathogens[1]. As inflammatory cells, they react to chemotactic signals and migrate to a wound or site of infection to protect the organism from pathogenic insults. Neutrophils are phagocytes that envelope and consume microbial invaders and on top of this capacity, they have an arsenal of antimicrobial properties at their disposal. They are highly cytotoxic cells full of degradative enzymes including proteases, hydrolases and nucleases[2]. In addition, neutrophils have the ability to produce and release reactive oxygen species, which can cross bacterial membranes damaging nucleic acids and proteins[3]. One of the most striking capabilities of these cells is to use their own DNA as a form of antimicrobial net in a process known as NETosis[4]. This is a programmed form of cell death where the neutrophil can release its chromatin to ensnare and kill larger microbes that are difficult to phagocytose such as fungal pathogens. As a collateral effect of their immune function, neutrophils secrete highly cytotoxic molecules that can irritate or damage surrounding tissues, contributing to inflammatory and autoimmune diseases[5]. However, it remains unknown to what extent aberrations in neutrophil function can drive immune disorders[6,7]. Pinpointing the causal cell types in these disorders is further complicated by the known crosstalk between neutrophils and other immune cell types such as B and T lymphocytes through cytokine messaging and direct interactions[8].

Genome-wide association studies (GWAS) are powerful tools for identifying the genetic determinants of complex traits and diseases, including autoimmunity. However, GWAS are cell type-agnostic and typically implicate non-coding variants that require further interpretation to be actionable. Causal GWAS variants preferentially localise to DNA regulatory elements such as enhancers, which bind transcription factors (TFs) to activate and fine-tune target gene expression[9,10]. Enhancers are often located large distances (up to megabases) away from their target genes and typically exert their effects via direct DNA looping contacts with their target gene promoters. Therefore, integrating GWAS results with the readouts of genome conformation and epigenome function is instrumental for identifying the causal cell types and molecular mechanisms behind GWAS signals. For example, enhancer activity is associated with specific patterns of chromatin accessibility and histone modifications (such as histone H3 lysine 4 methylation and lysine 27 acetylation), and we and others have previously shown that GWAS variants can associate with allele-specific imbalances in the magnitude of these readouts that potentially mediate the variants' expression effects[11–14]. The allelic effects of non-coding GWAS variants on gene expression[15], as well as their three-dimensional (3D) contacts with gene promoters determined using high-resolution chromosomal architecture profiling, enable the identification of target genes at GWAS traits[16,17].

PU.1 (encoded by *SPI1* gene) is a master TF controlling myeloid development[18]. PU.1 deficiency has profound effects on neutrophil maturation and function[19]. PU.1 is known to bind multitudes of enhancer elements in myeloid cells[20]. As a pioneer TF, PU.1 is able to bind these regions in a repressive chromatin state and activate them via the recruitment of other factors[21], including core chromatin activators and TFs such as C/EBPβ, which is expressed throughout neutrophil development[22]. PU.1 is required for modulating the response of mouse neutrophils to infection[23], but the emerging evidence for a role of this factor in complex diseases is currently limited to other cell types. For example, a common genetic variant within the *SPI1* gene was found to be associated with Alzheimer's disease, with microglial cells proposed as the likely causal cell type[24].

Here, to study the effects of PU.1 on immune function in human primary neutrophils, we use natural genetic variation across humans as a high-throughput "perturbation experiment" to identify PU.1 binding events in neutrophils that are subject to genetic control. We show that variants associated with PU.1 differential binding (PU.1 tfQTLs) are enriched for genetic associations with cell count and susceptibility to multiple autoimmune and inflammatory diseases. Integrating this information with other multi-individual genomic readouts, we show that the effects of genetically determined variation in PU.1 binding are likely propagated through the local chromatin state and 3D enhancer-promoter contacts to control the expression of disease-relevant genes. Jointly, these results provide evidence for a functional role of PU.1 in neutrophils in immune disease aetiology and provide mechanistic insights into the effects of non-coding genetic variation associated with healthy and pathological immune traits.

## Results

**A multi-individual atlas of PU.1 binding and associated genomic readouts.** We generated PU.1 chromatin immunoprecipitation sequencing (ChIP-seq) data from CD66b+CD16+ neutrophils isolated from fully genotyped individuals forming part of the BLUEPRINT project cohort[14,25], with data from a total of 93 donors passing quality control. In addition, we produced multi-individual data for two histone modifications, the active mark H3K4me3 and the Polycomb-associated inhibitory mark H3K27me3 in neutrophils isolated from the same donor cohort ($n = 107$ and $n = 106$, respectively). We also profiled the binding of the key PU.1 partner TF C/EBPβ and of the architectural protein CTCF in neutrophils ($n = 22$ and $n = 30$, respectively), as well as the binding of the same TFs in classical monocytes (PU.1, $n = 10$; C/EBPβ, $n = 9$; CTCF, $n = 4$) (Fig. 1a, Supplementary Fig. 1a, b and Supplementary Data 1). We identified bound regions (peaks) in ChIP-seq data using established algorithms (see "Methods") and confirmed data quality by assessing the fraction of reads in peaks (FRiP), as well as using principal component analysis and other approaches (Supplementary Fig. 1c–e and Supplementary Fig. 2a, b). Finally, we obtained high-resolution chromosomal interaction profiles of gene promoters in six individuals in neutrophils and monocytes by combining newly generated and published[16] Promoter Capture Hi-C (PCHi-C) datasets (Fig. 1a, Supplementary Fig. 3a–c).

**Genetic determinants of PU.1 binding in neutrophils.** To study genetically determined variation in PU.1 binding to DNA, we identified 36,530 high-confidence PU.1 peaks across the 93 individuals (Methods) and used normalised read counts at peak regions to determine TF quantitative trait loci associated with PU.1 binding (PU.1 tfQTLs; "Methods"). We detected 1868 independent PU.1 binding QTLs (linkage disequilibrium [LD] $r^2 \geq 0.8$; global false discovery rate [gFDR] <0.05; Supplementary Fig. 2c–e and Supplementary Data 2).

Lead PU.1 tfQTL SNPs showed a bimodal distribution of distances to their respective differential binding peaks (Fig. 1b, Supplementary Data 3), with just over half of them (55%, 1,036/1,868) mapping proximally from the peak edge (<2.5 kb; median distance 264 bp), and the remaining SNPs (45%; 995/1,868) localising more distally (2.5 kb-1 Mb, median distance 23 kb). tfQTL effect sizes were stronger for proximal compared to distal variants (Welch two-sided *t*-test $p = 2.2 \times 10^{-16}$, Fig. 1c) as previously observed in cultured lymphoblastoid cells[11,12]. We further validated a subset of the detected tfQTLs using allele-specific association analysis[26] ("Methods"), which confirmed a significant allelic imbalance for the majority of the tested peaks

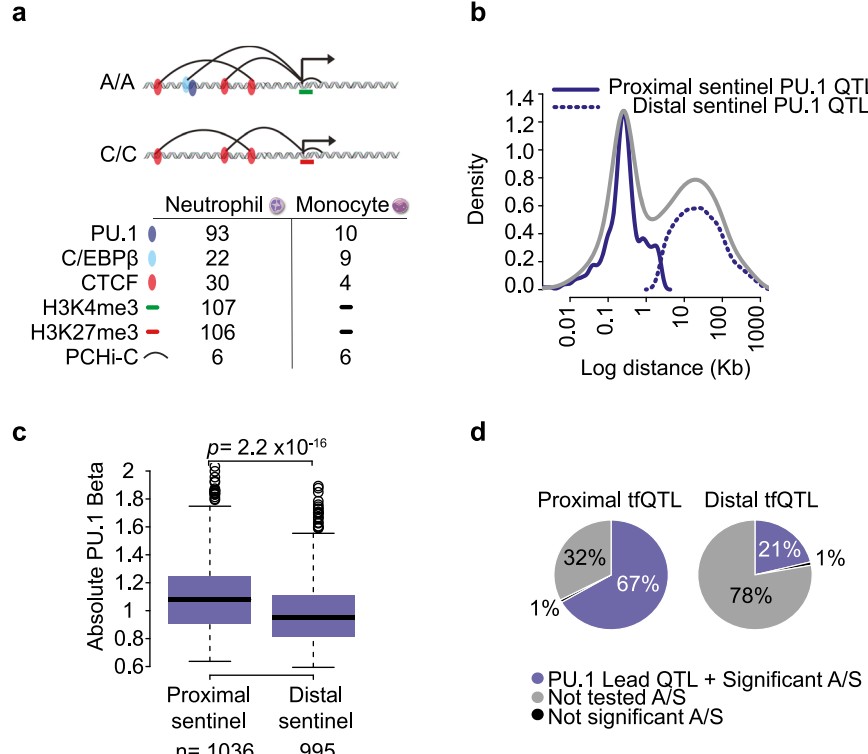

**Fig. 1 Properties PU.1 transcription factor QTLs. a** Summary of molecular traits generated as part of this study. **b** Density plot displaying the distance between sentinel SNPs and their associated PU.1 peaks. The bimodal distribution (grey) can be further subdivided into proximal (solid navy, <2.5 kb) and distal SNP effects (dotted, >2.5 kb). **c** Boxplot of absolute PU.1 tfQTL effect sizes (beta). Proximal PU.1 tfQTLs exhibit larger effect sizes compared to distal tfQTLs (Welch two-sided *t*-test). Box plots show the medians (centre lines) and the 25th and 75th percentiles (box edges), with whiskers extending to 1.5 times the interquartile range. *n* = the number of PU.1 tfQTL in each category. **d** Proportion of significant tfQTL SNPs with significant allele-specific (A/S) binding. Peaks without suitable heterozygous SNPs were not tested (grey).

(98.8% and 95.5% for peaks associated with proximal and distal variants respectively; Fig. 1d).

PU.1 tfQTLs associate with differential binding of other TFs in a distance-dependent manner. PU.1 binding to DNA leads to extensive chromatin remodelling[27] and redistribution of partner co-factors[28]. C/EBPβ is upregulated throughout neutrophil terminal differentiation[22] and has been shown to co-occupy myeloid enhancers at thousands of PU.1 bound sites[20]. In contrast, the constitutively expressed CTCF is known to play a role in gene regulation by anchoring chromatin interactions[29], but is not known to functionally associate with PU.1.

We identified 18,862 C/EBPβ and 22,197 CTCF filtered peaks from the datasets combined across individuals. We performed tfQTL analysis as before, identifying 427 C/EBPβ and 769 CTCF putative tfQTLs that reached a nominal *p* value threshold ($p \leq 1 \times 10^{-5}$; Supplementary Data 2). To assess the shared genetic effects on the binding of PU.1 and these two TFs *in cis*, we focused on proximal PU.1 tfQTLs and assessed the effect size on the nearest C/EBPβ and CTCF QTLs ($p \leq 1$) with respect to the lead SNP. PU.1 tfQTLs showed decreasing effects on C/EBPβ binding with increasing distance (linear regression $p = 2.2 \times 10^{-16}$), while no distance dependence on CTCF binding was observed ($p = 0.113$) (Fig. 2a, b). Indeed, significant C/EBPβ QTLs ($p \leq 1 \times 10^{-5}$, $n = 92$) were located proximal to the lead PU.1 tfQTL SNP, whereas shared significant CTCF tfQTLs ($p \leq 1 \times 10^{-5}$, $n = 135$) were located distally (Fig. 2c). These results confirm that PU.1 and C/EBPβ bind collaboratively and suggest long-range genetic effects on CTCF binding.

**Distal PU.1 binding variants are predominantly cell type specific.** We asked to what extent the identified PU.1 tfQTLs were specific to neutrophils. To this end, we additionally generated PU.1-binding maps in primary (CD14+CD16−) monocytes isolated from ten BLUEPRINT donors, five of whom were also part of the cohort used to profile PU.1 binding in neutrophils. The absolute majority (93%) of the PU.1-binding peaks associated with a tfQTL in neutrophils were also detected in monocytes (Supplementary Fig. 4a). The low number of individuals precluded an independent PU.1 tfQTL analysis in monocytes. Therefore, to assess coordination of genetic effects at PU.1-binding sites across cell types, we assessed the binding signal at shared PU.1 and C/EBPβ-binding sites in monocytes with individuals stratified by lead PU.1 tfQTL genotype. We found that monocytes displayed genotype associated changes in the strength of binding at proximal SNPs (linear regression $p = 3 \times 10^{-9}$) as observed in neutrophils ($p = 2 \times 10^{-13}$), compatible with shared genetic effects between the two cell types. However, the same was not true for distal SNPs (neutrophils $p = 4 \times 10^{-7}$, monocytes $p = 0.793$; Fig. 2d). This data suggest that proximal tfQTL variants in related cell types operate under a similar cis-regulatory mechanism, whereas long-range allelic control decays with distance and distal QTLs exhibit increased cell type specificity[30].

**PU.1 tfQTLs associate with local chromatin state.** Given that PU.1 is a pioneer factor, we expect the allelic effects of its binding to influence the local chromatin state. To verify this, we detected histone modification-associated QTLs (hQTLs) for the active H3K4me3 and repressive H3K27me3 histone marks in neutrophils and overlapped these data with PU.1 tfQTLs, identifying 621 and 367 shared PU.1 tfQTL/hQTL variants for these two marks, respectively (Supplementary Data 2). Variant effects on PU.1 binding and H3K4me3 tended to be in the same direction,

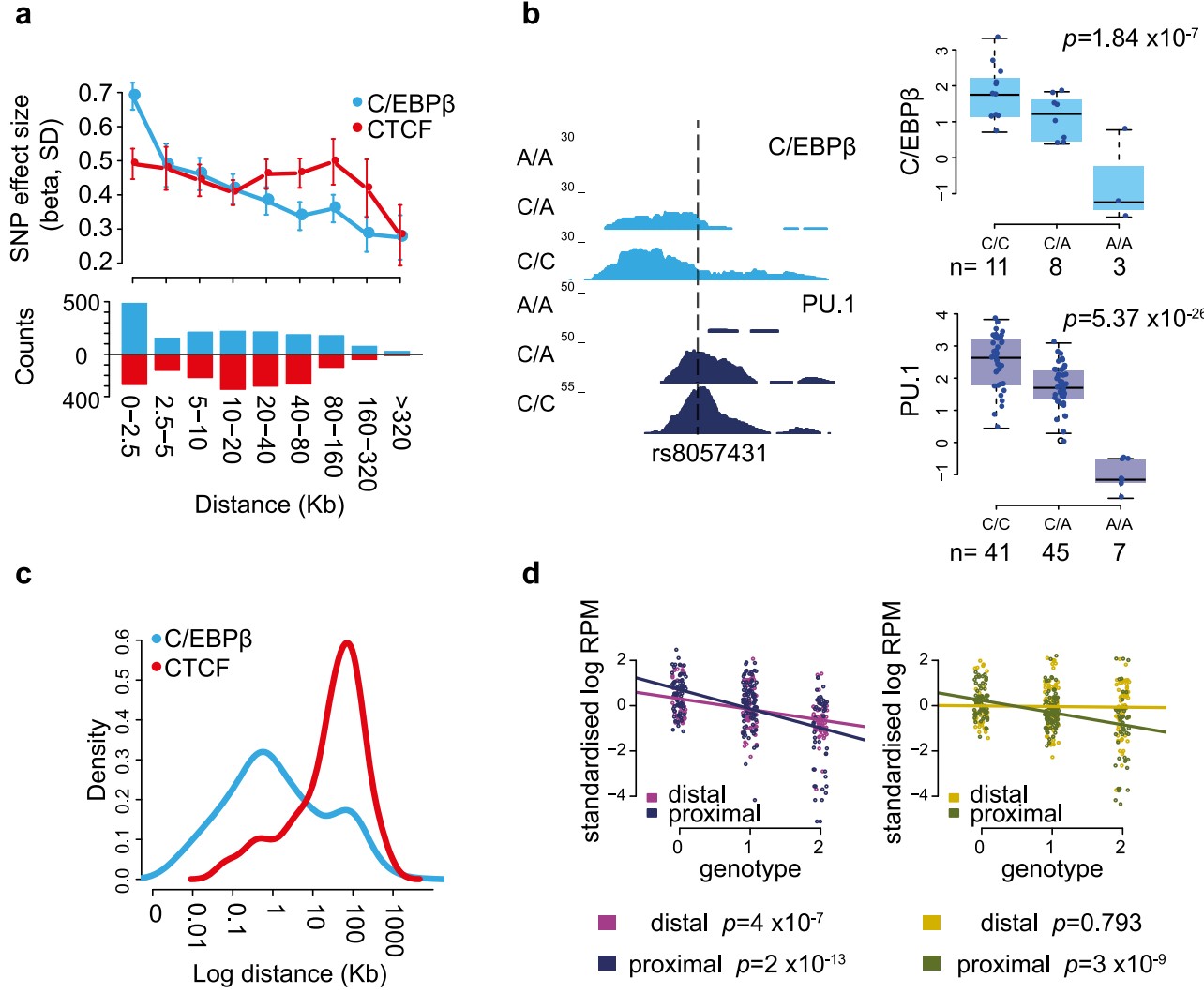

**Fig. 2 Effect of PU.1 SNPs on second transcription factor binding. a** Mean effect size (95% confidence intervals) for association of proximal sentinel PU.1 SNPs with the nearest C/EBPβ (light blue) and CTCF (red) binding site. The effect size decreases with distance for C/EBPβ (linear model $p < 2.2 \times 10^{-16}$) but not for CTCF (linear model $p = 0.113$). Beneath: bar chart of number of peaks included in each distance bin. **b** Genome browser shot of an illustrative example of tfQTL, where SNP rs8057431 (dashed line) alters a PU.1 motif and is associated with a disruption in binding of both PU.1 and C/EBPβ. With signal box plots created from all individuals segregated by genotype. Box plots show the medians (centre lines) and the 25th and 75th percentiles (box edges), with whiskers extending to 1.5 times the interquartile range. $p$ values were obtained by fitting linear mixed models implemented in LIMIX. $n$ = the number of independent donors. **c** Density plot displaying distance of lead proximal PU.1 SNP to the nearest shared association ($p < 10^{-5}$) C/EBPβ (light blue) or CTCF (red). **d** Transcription factor binding intensity for PU.1 ($y$ axis Log RPMs) at shared PU.1 and C/EBPβ tfQTL from five matched individuals in two cell types Neutrophils (left panel) and Monocytes (right panel). Binding sites were segregated by donor genotype ($x$-axis). tfQTL are categorised as either proximal or distal (33 distal and 43 proximal) and linear models were fitted separately for proximal and distal sites.

and in the opposite direction for PU.1 and H3K27me3 (Fig. 3a), consistent with the predominantly activating role of PU.1 at these loci. However, a significant minority ($n = 131$) of PU.1 tfQTLs showed the same direction of effect on PU.1 binding and the levels of the H3K27me3 repressive mark, suggesting a role for PU.1 acting as a repressor[23]. We additionally took advantage of the previously published hQTL data for the enhancer-associated histone marks H3K4me1 and H3K27ac[14]. In total, 808 H3K4me1 and 946 H3K27ac lead hQTL SNPs overlapped ($r^2 \geq 0.8$) PU.1 tfQTLs. Using the pi1 statistic[31], we found evidence of sharing between PU.1 tfQTLs with hQTLs in both neutrophils and monocytes (pi1(H3K27ac) = 0.73-0.76, and pi1(H3K4me1) = 0.76-0.80). Sharing between neutrophil PU.1 tfQTLs and hQTLs detected in CD4 naive T cells was lower (pi1$_{H3K27ac}$ = 0.36-0.72, and pi1$_{H3K4me1}$ = 0.30-0.79; Supplementary Fig. 4b), compatible with PU.1 being expressed at a lower level and having distinct

functions in lymphoid cells (Supplementary Fig. 4c)[32]. Notably, H3K27ac peaks co-occupied by PU.1 and C/EBPβ displayed greater hQTL effect sizes compared to peaks bound by PU.1 alone (Welch two-sided $t$-test $p = 7.28 \times 10^{-7}$; Fig. 3b), suggesting stronger genetic effects for enhancers at co-occupied sites in neutrophils. When compared with monocytes, neutrophil-specific PU.1 and C/EBPβ binding associated with neutrophil-specific chromatin activity (Supplementary Figs. 5 and 6). Collectively, these results show that genetically determined differences in PU.1 binding associate with differential chromatin states in a tissue-specific manner.

We next assessed the distance between the PU.1 and histone mark peaks for each shared tfQTL-hQTL genetic association. As previously observed in lymphoblastoid cell lines (LCLs)[11], there was a pronounced bimodal distribution of distances between PU.1 binding peaks and the locations of H3K27ac and H3K4me3

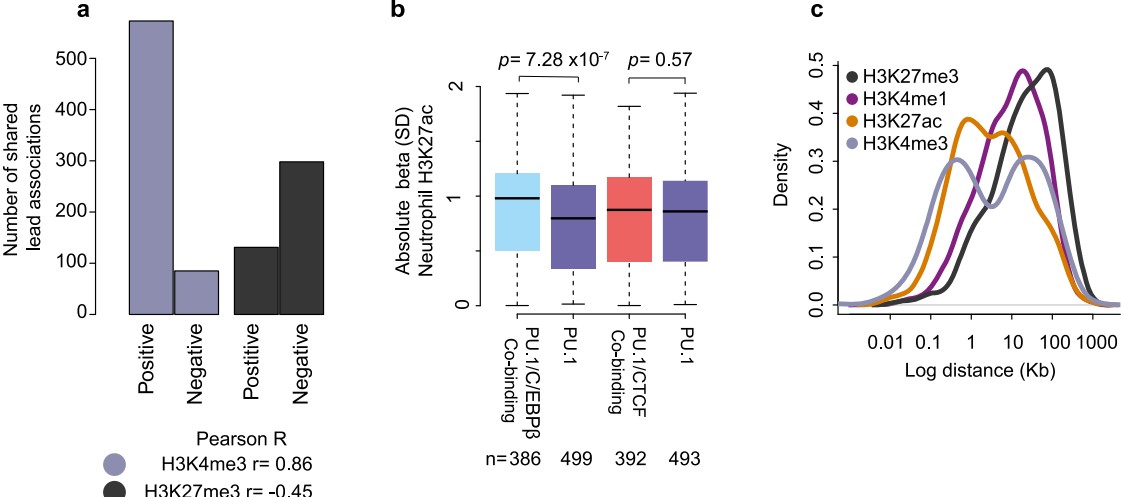

**Fig. 3 Neutrophil PU.1 tfQTLs and their association with chromatin state. a** Bar plot displaying the number of shared associations ($r_2 \geq 0.8$) between tfQTL and histone QTL. Bar plot is split into two categories, those cases where the beta between phenotypes are positively correlated, i.e., an allele leads to a gain (an increase in signal) in both phenotypes. Or negatively correlated, where an allele associates with a gain in one phenotype and a reduction in the second phenotype. The Pearson correlation between the betas is shown beneath. **b** Boxplot of absolute beta for H3K27ac neutrophil QTL (no significance threshold) for proximal lead PU.1 SNPs, differentiating H3K27ac regions that are or are not marked by C/EBPβ and/or CTCF binding. Box plots show the medians (centre lines) and the 25th and 75th percentiles (box edges), with whiskers extending to 1.5 times the interquartile range. $n =$ the number of PU.1 tfQTL that intersect H3K27ac hQTL. $p$ value displayed above is a Welch two-sided $t$-test. **c** Density distribution plot of log distance between lead PU.1 tfQTL SNPs and shared ($r^2 \geq 0.8$) histone QTL in neutrophils.

marks (Fig. 3c), with around a half of PU.1 peaks localising to less than 1 kb away from the respective H3K27ac and H3K4me3 peaks, and others mapping 10–100 kb from them. Given that H3K4me3 is associated with active promoters, this observation highlights the potential long-range regulatory effects of PU.1 binding to distal DNA elements on promoter activity.

**Allelic effects of PU.1 on enhancer-promoter contacts**. To further explore the long-range effects of PU.1 binding, we utilised our high-resolution PCHi-C data in neutrophils and monocytes from six donors in total. We detected ~295,000 Promoter Interacting Regions (PIRs) in total across neutrophils and monocytes (CHiCAGO score > 5)[33], ~110,000 of which were detectable in both cell types. We next explored the extent to which genetic variation affecting PU.1 binding associates with differential promoter-enhancer contacts. We employed an allele-specific strategy ("Methods") to identify heterozygous sites within PIRs that exhibited allelic imbalance at PCHi-C contacts (Supplementary Fig. 7a-b). We found an enrichment of tfQTLs within the ~14,000 heterozygous SNPs within PIRs that displayed evidence of allelic bias in both neutrophils and monocytes (Fig. 4a). Notably, the same was true for the hQTLs including the Polycomb-associated inhibitory mark H3K27me3, consistent with a role of Polycomb repressive complexes in shaping regulatory chromatin architecture[34] (Supplementary Fig. 7c–e).

**Effects of PU.1 tfQTLs on distal gene expression**. To assess the effects of PU.1 tfQTLs on distal gene expression, we initially took advantage of expression QTL (eQTL) data generated from the same individuals[14], detecting 609 shared PU.1 tfQTLs/eQTLs ([LD] $r^2 \geq 0.8$), with a median distance from tfQTL to transcription start site of 56.5 kb. Notably, for 128 of 905 eQTL genes PU.1 binding was negatively correlated with gene expression, reinforcing the role of PU.1 as a dual transcriptional activator and repressor[23]. An interesting example of a repressive PU.1 tfQTL was rs2149092, associated with *RNASET2* gene expression, and

previously implicated in the pathogenesis of inflammatory bowel disease (IBD)[35] (Supplementary Fig. 8a–c).

Distal eQTL signals do not discriminate between direct and indirect effects. Therefore, to investigate the direct role of PU.1 in long-range gene control, we again turned to our PCHi-C data. PIRs enriched in neutrophil specific PU.1-binding sites (presented in Supplementary Fig. 6a) and enhancer-associated H3K4me1/H3K27ac marks were positively correlated with the level of expression of the genes they contacted in neutrophils (Fig. 4b), similarly as shown for other cell types[36]. In contrast, CTCF binding at PIRs did not correlate with target gene expression (Fig. 4b), as expected given the constitutive nature of many CTCF-mediated chromosomal interactions. In addition, we observed no association with monocyte-specific PU.1 binding, promoter interactions and gene expression levels (Fig. 4b). Finally, we compared the effects of distal PU.1 tfQTLs/eQTLs (>25 kb from TSS) mapping within and outside PIRs. PU.1 tfQTLs mapping to PIRs showed significantly larger effects on the expression of the genes they contacted compared with distance-matched SNPs that did not map to a PIR (two-sided Fisher's exact test $p < 2 \times 10^{-16}$; Fig. 4c), demonstrating that the physical enhancer-promoter contacts mediate the transcriptional effects of PU.1 binding.

An example of a PU.1 tfQTL SNP showing allelic imbalance affecting promoter-enhancer connectivity and gene expression is rs519989, an eQTL of the *LRRC8C* gene (Fig. 4d–g). *LRRC8C* encodes a volume-regulated anion channel subunit upregulated during terminal differentiation of neutrophils[37]. This and other loci (including Supplementary Fig. 8c) demonstrate coordinated genetic influences on PU.1 binding, chromatin activity and the formation of promoter interactions in the regulation of neutrophil gene expression.

**PU.1 differential binding underpins disease-associated variants**. Genome-wide association studies have been used to identify regions of the genome associated with phenotypic traits and disease. In addition, TF-binding maps and epigenetic profiling has been shown to assist in pinpointing the causal variants within

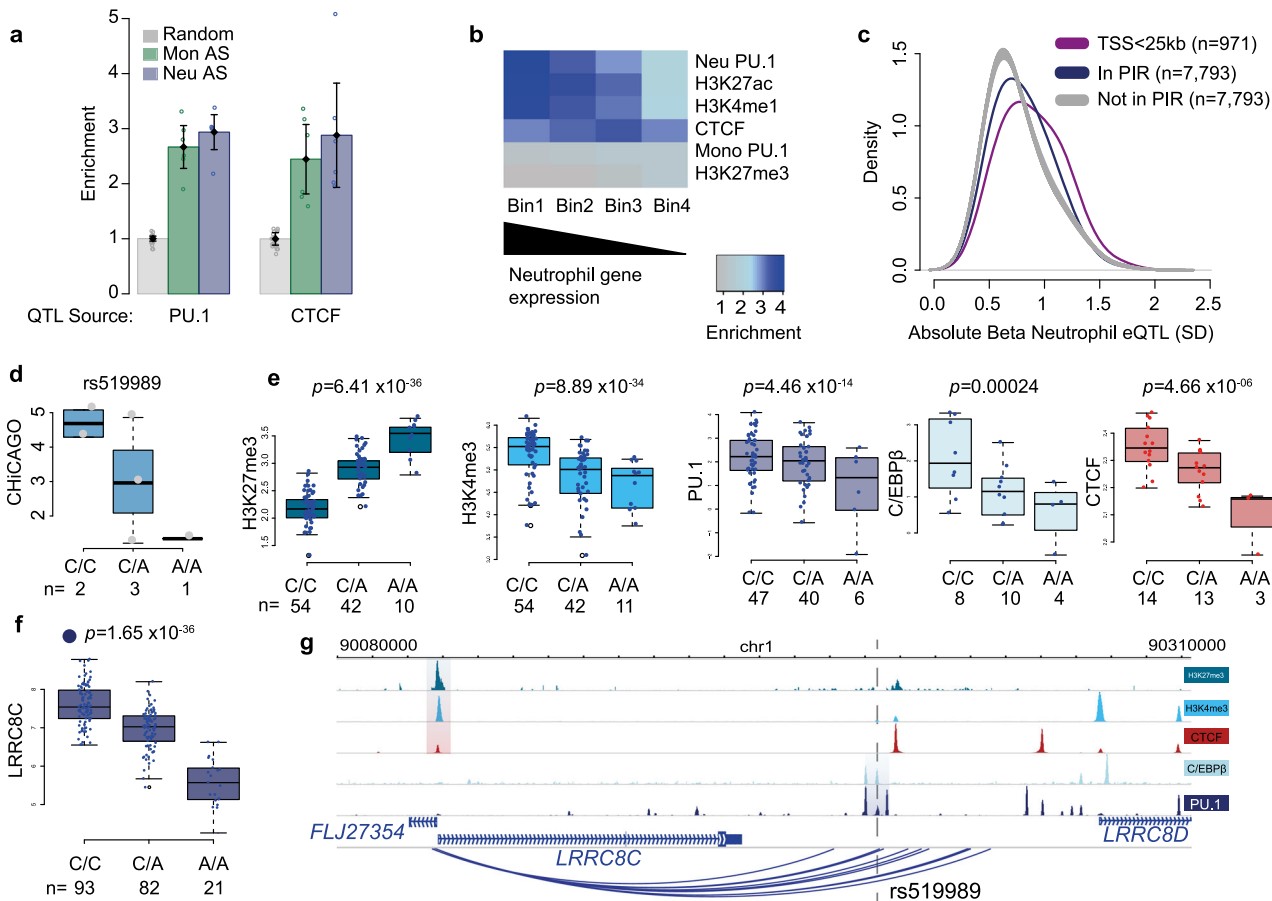

**Fig. 4 tfQTLs perturb gene expression through altered chromatin state. a** Enrichment of significant tfQTLs (PU.1 and CTCF; Fisher's exact test $p < 1 \times 10^{-5}$) in PIRs of both neutrophils and monocytes. The bars represent the mean and the error bars the 95% confidence interval. **b** Heat map showing enrichment of transcription factor or histone modified regions intersecting PIRs, whereby PIRs were ranked into four bins based on the gene expression of the connected baited genes in neutrophils. **c** Density plot of gene expression QTL Beta value for neutrophil PU.1 SNPs within PIRs (navy) versus distance-matched significant SNPs not in PIRs (grey) (two-sided Fisher's exact test $p < 2 \times 10^{-16}$), and distribution of beta values for SNPs within <25 kb of transcription start site (purple). The SNPs that are not in PIRs are also significant PU.1 tfQTLs and eQTLs (linear model $p < 1 \times 10^{-5}$ cut off). **d** CHiCAGO scores for the PIR at tfQTL, segregated by donor genotype for rs519989. *n* the number of individual donors. **e** Signal box plots with donors separated by genotype for rs519989 for five molecular traits, **f** boxplots displaying RNA level for *LRRC8C* gene segregated by donor genotype for SNP rs519989. $n =$ the number of individual donors. Box plots show the medians (centre lines) and the 25th and 75th percentiles (box edges), with whiskers extending to 1.5 times the interquartile range. *p* values were obtained by fitting linear mixed models implemented in LIMIX. **g** Genome browser view of region around *LRRC8C* gene, QTL regions for each molecular trait are highlighted. Dashed line depicts the location of rs519989.

these regions and implicated mechanisms associated with complex traits. To look for evidence of a functional relationship between our identified transcription factor binding sites and the shaping of myeloid traits, we utilised available summary statistics for monocyte and neutrophil counts[38]. Using GARFIELD (GWAS Analysis of Regulatory or Functional Information Enrichment with LD correction)[39], we observed enrichment of variants associated with full blood count (FBC) measurements within PU.1, C/EBPβ and, interestingly, CTCF bound regions of the neutrophil genome (Fig. 5a). This analysis is suggestive of a relationship between regulatory regions of the neutrophil genome and genetic variants associated with myeloid traits. To further investigate whether the identified tfQTLs potentially influence human health, we took advantage of the diverse range of reported traits in 361,194 individuals from the UK Biobank study. We obtained summary statistics from 24 GWAS for autoimmune, allergy and infection associated traits generated from the UK Biobank resource (http://www.nealelab.is/uk-biobank/). From these we observed inflation of *p* values for PU.1 tfQTLs within GWAS for hay fever, asthma, rheumatoid arthritis (RA) and ulcerative colitis (UC), but perhaps somewhat

surprisingly, not at those associated with infections such *Clostridium difficile* (Supplementary Fig. 9).

To formally test the overlap of PU.1 tfQTLs and haematological traits and diseases we accessed summary statistics from the aforementioned GWAS on FBC traits[38] and published autoimmune diseases[40–45]. Applying colocalisation analysis[46,47] revealed 26 proximal and 53 distal tfQTLs that shared a genetic signal (posterior probability [PP] > 0.9) with at least one GWAS locus (Fig. 5b, Supplementary Data 4). We detected an overlapping signal at 4 loci (rs3784789, rs13089544, rs9262174 and rs13035725) between tfQTL, FBC and disease GWAS, 3 of which could be attributed to a lead SNP proximal to a PU.1-binding site. To determine the putative target genes underpinning PU.1-mediated disease associations, we integrated PCHi-C and eQTL data in neutrophils. Overall, 27 high-confidence target genes interacting with eQTL loci colocalised with GWAS summary statistics (Supplementary Data 5).

An example of a PU.1 tfQTL associated with myeloid traits is the rs791357_C variant associated with decreased neutrophil and monocyte cell counts. PCHi-C data show that this region is highly connected to the *CPEB4* gene in both neutrophils and monocytes

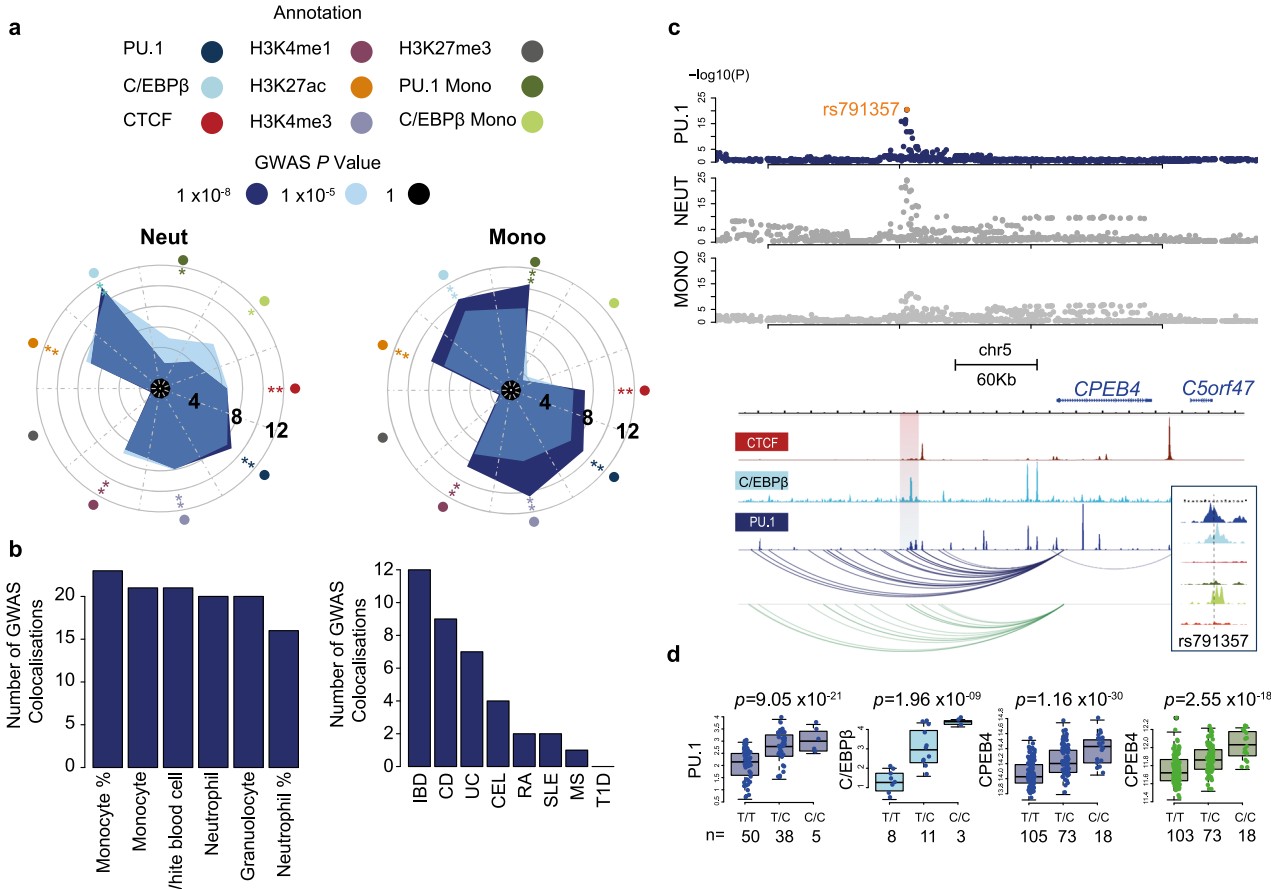

**Fig. 5 tfQTLs influence cellular phenotype and disease. a** Circos plots displaying fold-enrichment of GWAS loci for neutrophil count and monocyte count within neutrophil regions marked by TF binding, modified histones and PU.1 and C/EBPβ specific binding sites in monocytes. Radial grid lines for published GWAS *p* values, asterisk denotes significance of enrichment for annotation tested at each GWAS *p* value cut off. **b** Bar plot displaying the number of GWAS loci that colocalise with PU.1 tfQTL for both full blood count phenotypes (left) and autoimmune disease (right). **c** Example of colocalised signal for sentinel SNP rs791357 tfQTL with a shared association for both neutrophil and monocyte count traits. Manhattan plots showing −log10(*P*) value for shared SNPs from published GWAS for neutrophil and monocyte counts obtained by fitting linear models, and PU.1 tfQTL (navy) obtained from LIMIX. With genome browser visualisation of locus for CTCF, C/EBPβ and PU.1 binding shown beneath. The top associated peak is highlighted by the shaded area. In addition, PCHi-C data show that this region is highly connected to the enhancer region in both neutrophils (blue) and monocytes (green). **d** Boxplot for TF and RNA signal segregated by donor genotype. PU.1 (navy), CEBPβ (light blue), *CPEB4* gene expression neutrophil (navy) and *CPEB4* gene expression monocyte (olive green). Box plots show the medians (centre lines) and the twenty-fifth and seventy-fifth percentiles (box edges), with whiskers extending to 1.5 times the interquartile range. *p* values were obtained by fitting linear mixed models implemented in LIMIX. *n* = the number of individual donors.

(Fig. 5c-d). CPEB4 is a cytoplasmic polyadenylation protein which binds to a recognition sequence in the poly-A tail of mRNAs to regulate translation[48]. The SNP rs791357 is a proximal tfQTL for PU.1 ($p = 9.05 \times 10^{-21}$) and C/EBPβ ($p = 1.963 \times 10^{-9}$), an hQTL for H3K4me3 ($p = 1.98 \times 10^{-17}$) and H3K27ac ($p = 1.41 \times 10^{-33}$) and an eQTL for *CPEB4* ($p = 1.16 \times 10^{-30}$) in neutrophils. Similar sharing of PU.1 and C/EBPβ binding sites was observed in monocytes with hQTL (H3K27ac $p = 8.14 \times 10^{-26}$) and eQTL for *CPEB4* ($p = 2.55 \times 10^{-18}$).

The aforementioned *RNASET2* gene (Supplementary Fig. 8) has been posited previously as an IBD risk-associated gene in CD4[+] T cells[35]. Moreover, it was identified as a target of TL1A, a mediator of inflammation in the gut. In our neutrophil QTL data we identified rs2149092 as the nearest SNP to the PU.1 tfQTL (chr6:167371914-167373868), which is in LD($r^2 = 1$) with the lead reported GWAS SNP rs1819333. We confirmed a colocalisation between this PU.1 tfQTL and GWAS for IBD and Crohn's disease (CD) ([PP] = 1). In addition, this tfQTL has a significant association with H3K4me3 ($p = 6.11 \times 10^{-17}$) at the

promoter of the *RNASET2* gene, with the expression of this gene itself ($p = 5.75 \times 10^{-61}$) and with a CTCF binding site ($p = 8.82 \times 10^{-6}$) in the gene body. Interestingly, there was an association with H3K27me3 ($p = 3.71 \times 10^{-22}$) at the promoter region and with expression of the *RPS6KA2* gene ($p = 1.72 \times 10^{-23}$), which interacts with the lead SNP. The *RPS6KA2* gene has been implicated as an IBD risk-associated gene in an epigenome-wide association study examining differentially methylated regions in IBD patients from whole blood[49]. Here we identify a genetic link suggesting that PU.1 is playing a pivotal role in the underlying pathogenicity of IBD at this locus.

We further observed several examples of PU.1-mediated gene regulation that may point to neutrophils as being the effector cell type in the pathogenesis of inflammatory disease. The GWAS colocalised loci include five shared tfQTL/eQTL in neutrophils without a significant eQTL in monocytes (rs6658646, rs3768421, rs77120970, rs16895831 and rs3817465). The latter is an eQTL for the autoimmune associated gene *ILRAP18* ($p = 2.16 \times 10^{-27}$ in neutrophils and $p = 0.0457$ in monocytes). In

addition we identified loci colocalised with genes that have been previously implicated in neutrophil biology. For example CXCR2 is involved in neutrophil migration and has been implicated in inflammatory disease, through the regulation of neutrophil extracellular trap formation in patients suffering from chronic obstructive pulmonary disease[50]. We observed a colocalised signal for PU.1 tfQTL, rs13035725 ($p = 2.44 \times 10^{-10}$) with IBD and UC. The same SNP is also significantly associated with a CTCF tfQTL ($p = 5.52 \times 10^{-6}$) and expression of several genes in neutrophils, including CXCR2 ($p = 4.25 \times 10^{-13}$), ARPC2 ($p = 3.70 \times 10^{-20}$), AAMP ($p = 1.02 \times 10^{-8}$) and CXCR1 ($p = 1.23 \times 10^{-6}$), all of which were weakly or not significant in monocytes. The region also displays high connectivity in the neutrophil PCHi-C data (CHiCAGO score > 5), but is depleted of significant interactions in monocytes (Supplementary Fig. 10a, b).

The PLCL1 locus is another interesting example. The region harbours variants associated with susceptibility to several diseases including CD[51], systemic lupus erythematosus[52] and allergy[53]. PLCL1 encodes the phospholipase C epsilon or phospholipase C like I (inactive) signalling protein that has been shown to be involved in receptor turnover but also inhibiting integrin activity[54], suggesting a role in the regulation of cell trafficking. We detected a colocalised signal with our PU.1 tfQTL and GWAS for CD, with our lead tfQTL rs9712275 ($p = 2.11 \times 10^{-18}$) being in high LD ($r^2 > 0.9$) with the reported GWAS variant rs6738825. In this case, the tfQTL contacts the PLCL1 gene promoter in both neutrophils and monocytes and we observed evidence of allelic imbalance in both cell types (Supplementary Fig. 10c, d). The lead SNP rs9712275 is an eQTL for the PLCL1 gene in both neutrophils ($p = 2.86 \times 10^{-27}$) and monocytes ($p = 9.14 \times 10^{-36}$).

Finally, we used CATO (Contextual Analysis of TF Occupancy)[47] to identify collaborating factors that may be involved in mediating these traits at shared PU.1 tfQTL/GWAS loci. Colocalising SNPs were shown to affect binding recognition motifs for several PU.1 binding partners, including C/EBP, AP-1, ETS, CTF/NF-1, ATF/CREB and RUNX (Supplementary Fig. 11). These results highlight the likely role of PU.1 and its partners in mediating the functional effects of GWAS variants in neutrophils. Jointly, these findings demonstrate how genetically determined variation in PU.1 binding, and potentially other partners, underpins disease traits via altered chromatin states at regulatory regions and long-range chromosomal contacts.

## Discussion

In this study, we have leveraged common genetic variation as "nature's perturbation screen" to study the functional implications of the binding of key myeloid TF PU.1 in human primary neutrophils. We have identified thousands of DNA variants associated with altered PU.1 binding (both *in cis* and *in trans*), and show that they also associate with downstream transcriptional regulatory events, including local chromatin state, long-range enhancer-promoter contacts and gene expression. We further show that PU.1-binding variants underpin multiple GWAS variants for healthy and pathological immune traits, including autoimmunity, implicating PU.1 and neutrophils in the aetiology of these processes.

GWAS have been deployed successfully over the last 15 years to identify thousands of genetic loci associated with complex traits and disease. To facilitate the utility of this endeavour into tangible improvements in human health there remains a necessity to identify disease-relevant cell types, biological pathways and target genes. The epigenetic profiling of cell types implicated in disease aids in the mapping of disease causative genetic variants[10,55]. Our study adds to the emerging array of analyses using multi-individual readouts of the chromatin state and TF

binding to understand the function of GWAS variants[13]. Many of these studies, including multi-individual PU.1-binding profiling[11,56], were performed in LCLs. However, using disease-relevant primary cells provides appropriate cellular context and obviates in vitro artefacts, which is particularly important for neutrophils that are known to have pronounced differences in gene expression patterns from the established cell culture models[57]. Inclusion of a second primary myeloid cell type in our analysis (monocytes) has enabled pinpointing cell-type-specific effects of genetic variation, and revealing that they preferentially affect distal PU.1 tfQTLs.

The fact that large numbers of GWAS variants tag distal enhancers makes the identification of the target genes underpinning GWAS associations a challenging task. Insights from expression quantitative trait loci (eQTLs) and 3D chromosomal contacts between enhancers and promoters are proving to be instrumental for GWAS gene assignment[16,58]. Here we used neutrophil eQTL data from the BLUEPRINT project[14] and high-resolution PCHi-C data in primary neutrophils to link PU.1 tfQTLs and the GWAS variants they underpin with their target genes. PCHi-C data from six individual donors, which emcompasses the most detailed datasets to date, also enabled us to identify the potential allelic effects of promoter interactions associated with PU.1 differential binding. This analysis extends existing multi-individual studies using conventional Hi-C to high-resolution promoter interaction analysis[59,60]. Both our and the previous published analyses, however, used relatively small numbers of individuals for chromosomal architecture profiling, and further robust associations will likely be detected by future higher-powered studies.

PU.1 has long been known as the key regulator of myeloid development[61]. However, the role of this factor in mature neutrophils has remained less well explored. A recent study by Fischer et al. has shown that deletion of PU.1 in mouse mature neutrophils affects the expression of a multitude of genes involved in immune response and results in a hyperinflammatory phenotype following infection[23]. Our findings that human neutrophil PU.1 tfQTLs overlap eQTLs and autoimmune GWAS traits provide further evidence for the role of this TF in controlling the transcriptional state of mature neutrophils and point to potential clinical implications of its aberrant binding.

Our observation that PU.1 binding associates with both active and repressive chromatin marks, as well as both directly and inversely correlates with the effects of overlapping eQTLs, corroborates the evidence from Fischer et al. of the importance of PU.1 as a transcriptional repressor in the inflammatory context. Our results are also consistent with the established role of PU.1's upstream regulator RUNX3 in autoimmune diseases such as psoriatic arthritis[62]. TFs commonly act cooperatively[63], and as expected, many PU.1 binding variants have shared effects on the binding of its partner TF C/EBP. Cooperativity likely underpins the initially puzzling findings from complementary studies of other TFs, which showed that the majority of TF-binding variants are located outside the conventional TF motif[64]. Notably, the abundance of distal PU.1 binding variants, consistent with previous findings[11], suggests that such cooperativity may extend over considerable genomic distances, potentially through DNA looping[30]. The shared genetic effects that we observe on the binding of PU.1 and the key chromosomal architectural protein CTCF, in many cases with respect to sites mapping distally to each other, support this view.

Over the recent years, neutrophils have emerged as important regulators of autoimmune response. For example, neutrophils are the most abundant cell type found in the synovial fluid between joints of RA patients[65]. In IBD, neutrophils are known to damage the mucosal epithelium and release monocyte chemoattractants,

leading to an extended inflammatory episode in the gut[66]. Our findings that GWAS variants for autoimmune disease susceptibility affect the molecular phenotypes in primary neutrophils reinforce the role of this cell type in autoimmunity and suggest that neutrophils may, at least in some cases, play a driver role in the development of these conditions as opposed to merely being "support players" in autoimmune response.

Beyond the context of the present study, we release a large resource of molecular phenotypes associated with non-coding variants in primary immune cells that we envisage to be useful for the community to further decipher the mechanistic underpinnings of immune gene control and variant-disease associations, potentially used in conjunction with the emerging data on in vitro mutagenesis and CRISPR-targeted chromatin perturbation[67].

## Methods

**Peripheral adult blood collection**. Data generated in this study used donor samples which were collected as part of the previously described study[14]. Blood was obtained from donors who were members of the NIHR Cambridge BioResource (http://www.cambridgebioresource.org.uk/) with informed consent (Ethical oversight by NRES Committee East of England, Hertfordshire, Study title: A Blueprint of Blood Cells, REC 12/EE/0040) at the NHS Blood and Transplant, Cambridge. Donors were on average 55 years old (range 20–75 years old), with 46% of donors being male. A unit of whole blood (475 ml) was collected in 3.2% sodium citrate. An aliquot of this sample was collected in EDTA for genomic DNA purification. A FBC for all donors was obtained from an EDTA blood sample, collected in parallel with the whole blood unit, using a Sysmex Haematological analyser. The level of C-reactive protein (CRP), an inflammatory marker, was also measured in the sera of all individuals. All donors used for the collection had FBC and CRP parameters within the normal healthy range. Blood was processed within 4 h of collection.

**Isolation of cell subsets**. Samples were as those as described[14]. To obtain pure samples of 'classical' monocytes (CD14+CD16−) and neutrophils (CD66b+CD16+) we implemented a multi-step purification strategy. Whole blood was diluted 1:1 in a buffer of Dulbecco's phosphate buffered saline (PBS, Sigma) containing 13 mM sodium citrate tribasic dehydrate (Sigma) and 0.2% human serum albumin (HSA, PAA) and separated using an isotonic Percoll gradient of 1.078 g/ml (Fisher Scientific). Peripheral blood mononuclear cells were collected and washed twice with buffer, diluted to 25 million cells/ml and separated into two layers, a monocyte rich layer and a lymphocyte rich layer, using a Percoll gradient of 1.066 g/ml. Cells from each layer were washed in PBS (13 mM sodium citrate and 0.2% HSA) and subsets purified using an antibody/magnetic bead strategy. To purify monocytes, CD16+ cells were depleted from the monocyte rich layer using CD16 microbeads (Miltenyi) according to the manufacturer's instructions. Cells were washed in PBS (13 mM sodium citrate and 0.2% HSA) and CD14+ cells were positively selected using CD14 microbeads (Miltenyi). To purify neutrophils, the dense layer of cells from the 1.078 g/ml Percoll separation was lysed twice using an ammonium chloride buffer to remove erythrocytes. The resulting cells (including neutrophils and eosinophils) were washed and neutrophils positively selected using CD16 microbeads (Miltenyi) according to the manufacturer's instructions. The purity of each cell preparation was assessed by multicolour FACS using conjugated antibodies for CD14 (MφP9, BD Biosciences) and CD16 (B73.1 / leu11c, BD Biosciences) for monocytes, CD16 (VEP13, MACS, Miltenyi) and CD66b (BIRMA 17C, IBGRL-NHS) for neutrophils. Purity was on average 95% for monocytes and 98% for neutrophils.

**ChIP-sequencing**. Purified cells were fixed with 1% formaldehyde (Sigma) at a concentration of approximately 10 million cells/ml. Fixed cell preparations were washed and stored re-suspended in PBS at 4 °C prior to lysis and sonication. Sonication protocols were performed in a Diagenode PicoRuptor for 8 cycles of 30 s on, 30 s off in a 4 °C water cooler. Samples were checked for sonication efficiency using the criteria of 150–500 bp, by Agilent DNA bioanalyzer. ChIP-seq was carried out as previously described[68] all liquid handling steps were performed on an Agilent Bravo NGS. Protein A Dynabeads (Invitrogen) were coupled with 2.5 μg of antibody. Sonicated lysate (3–5 million cells) was then added to the bead/antibody mix and incubated at 4 °C overnight. ChIP-DNA bound beads were washed for ten repetitions in cold RIPA solution. Elution of DNA from beads at 65 °C for 5 h to reverse the cross linking process. Two microlitres RNase was added to ChIP-DNA and incubated at 37 °C for 30 min, followed by 2 μl of Proteinase K treatment at 55 °C for 1 h. 1:1.8 ratio of Ampure beads (Beckman Coulter, A63881) were added to the DNA followed by two cold 70% ethanol washes. ChIP-DNA was eluted in 50 μl elution buffer. Illumina sequencing libraries were prepared on a Beckman Fx liquid handling system. End-repair, A-tailing and paired-end adaptor ligation were performed using NEBnext reagents from New England Biolabs (E6000S), with purification using a 1:1 ratio of AMPure XP to sample between each reaction. Amplification of ChIP-DNA was performed using Kapa HiFi master mix (Kapa Biosystems KK2602), 18 cycles of PCR followed by a 0.7:1 Ampure XP clean-up. Antibodies for H3K4me3 (C15410003, lot A5051-001D), H3K27me3 (C15410195, lot A1811-001D), CTCF (C15410210, lot A2359-0010) were obtained from Diagenode, Liege, Belgium. Antibodies for PU.1 (sc-352x, lot B2415 and sc-22805x, lot D0609) and C/EBPβ (sc-150x, lot G1814) were obtained from Santa Cruz Biotechnology.

**Data processing and peak calling**. ChIP libraries were sequenced using Illumina HiSeq 2000 and HiSeq 2500 at 50 bp single end reads. Sequenced reads were aligned to reference genome (hg19) using BWA (bwa aln −q 15). Duplicate reads were marked using Picard MarkDuplicates (v1.103). Reads with mapping quality less than 15 (MAPQ 15) were removed (SAMtools v0.1.18). The fragment size L for each aligned bam was estimated using PhantomPeakQualTools vr18, which uses cross correlation of binned read counts between forward and reverse strands. To identify highly enriched genomic regions, we used MACS2[69] (v2.0.10.20131216, standard options) for peak calling with the estimated fragment size from PhantomPeakQualTools (--shiftsize = half fragment size), with narrow for PU.1, C/EBPβ, CTCF, H3K4me3 and broad flags set for H3K27me3. For background control ChIP input was created from merging random selected samples. Reads from 4 pools of 12 individuals for neutrophil input and 2 pools of 6 individuals for monocytes (Supplementary Data 1). Significant peaks were selected to be at 1% FDR or less.

**Data quality**. We removed ChIP samples that had a relative strand correlation (RSC) < 0.8 and normalised strand correlation (NSC) < 1.05[70]. We defined high-confidence data from those ChIP with RSC > 0.8 and NSC > 1.05. Otherwise, we used genome browser tracks to confirm visually a good ChIP and include it in the final data set. Supplementary Fig. 1 and Supplementary Data 1 shows quality control metrics and corresponding principal components, showing no batch effects after PEER correction using $K = 10$ factors.

**Normalised read count in the reference peak sets**. Consensus peak sets were constructed using dba.peakset function within DiffBind R (ver3.4.0) package. http://bioconductor.org/packages/release/bioc/vignettes/DiffBind/inst/doc/DiffBind.pdf[71]. For PU.1, H3K4me3 and H3K27me3, we set the minimum number of samples for a peak to be included in consensus to 3, for C/EBPβ, CTCF and monocyte samples minimum was set to 2. Sex chromosomes were not included in the QTL analysis. The reference peak set was filtered further for read counts as described below. Next, we generated a quantification signal of ChIP-seq for each donor. Here we only considered read counts under the peaks, as the regions outside peaks are more likely to be noise or background signal than true enrichment. For each donor, we generated a vector of log2 reads per million (log2RPM) per peak in the reference peak set by counting the number of overlapping reads under the peaks (BEDOPS v2.4.14 bedmap-count) and normalised the counts with the total number of reads in the library. We further filtered the reference peak set to only consider peaks with log2RPM > 0 in at least 50% of the donors in a given cell type, corrected for ten PEER factors and applied quantile normalisation across donors. For QTL calling with H3K27me3, two sets of summary statistics are provided on two separate signal matrices. In the first set H3K4me3 peak annotations were used in conjunction with H3K27me3 signal to enrich for poised promoter QTLs. In the second set broad called H3K27me3 peaks were divided into 2500 bp windows.

**Identification of PU.1 and C/EBPβ differential binding sites**. We used DiffBind version 1.12.0 with default EdgeR (3.8.3) option to identify peaks which were differentially bound between neutrophils and monocytes. For each cell type we used the six best quality samples and their peak sets for this analysis; PU.1: NS1509, NS1510, NS1511, NS1514, NS1516, NS1522, NS1463, NS1464, NS1554, NS1551, NS1437 and NS1490. C/EBPβ; NS1559, NS1565, NS1563, NS1558, NS1562, NS1566, NS585, NS743, NS791, NS729, NS717, and NS793. Quality control plots generated as part of this analysis include correlation between sample peak sets (Supplementary Fig. 5a) and PCA between samples using normalised counts within consensus peak set (Supplementary Fig. 5b). Differentially bound peaks were considered to those peaks present in at least three individuals with a minimum threefold difference (FDR < 0.05) in binding signal as cut off. The heat map visualisation of differentially bound regions Deeptools 2(Galaxy Version 3.3.2.0.1)[72].

**TF enrichments**. For determining enrichment of ChIP-seq regions of interest within PIRs we used regioneR (1.0.3)[73], which performs a statistical evaluation of two sets of genomic regions by permutation testing. We set to 50 permutations the randomisation of genomic regions to determine the null. Differentially expressed genes and gene expression counts. Gene expression counts and list of differentially expressed genes were available from Ecker et al[25].

**QTL mapping**. Cis-acting QTL mapping was done using the LIMIX package[74], available from github (https://github.com/PMBio/limix). We considered genetic variants mapping to within 1 Mb (on each side) of each tested feature (peak), and tested their association using linear regression. Models were fit on quantile-

normalised PEER residuals, also including a random effect term accounting for polygenic signal and sample relatedness (as in the variance component models above we used the realised relatedness matrix to capture sample relatedness). From the linear regression we obtained the effect size and $p$ value for each tested association. To correct for multiple hypothesis testing, we performed a two-step procedure[75]: first, we corrected for multiple testing across variants for each molecular outcome using Bonferroni correction and, second, we adjusted the obtained $p$ values for multiple testing across phenotypes within each layer using a the $Q$ value procedure[31], considered QTLs at a significance threshold of 5% FDR.

**PCHi-C.** Cells were isolated as described[16]. One donor was used for preparing each PCHi-C library. In total, 12 PCHi-C libraries were prepared, six using monocytes and six using neutrophils. Approximately $8 \times 10^7$ cells per library were resuspended in 30.625 ml of DMEM supplemented with 10% FBS, and 4.375 ml of formaldehyde was added (16% stock solution; 2% final concentration). The fixation reaction continued for 10 min at room temperature with mixing and was then quenched by the addition of 5 ml of 1 M glycine (125 mM final concentration). Cells were incubated at room temperature for 5 min and then on ice for 15 min. Cells were pelleted by centrifugation at 400 g for 10 min at 4 °C, and the supernatant was discarded. The pellet was washed briefly in cold PBS, and samples were centrifuged again to pellet the cells. The supernatant was removed, and the cell pellets were flash frozen in liquid nitrogen and stored at −80 °C. HiC library generation was carried with in-nucleus ligation as described previously[76]. Chromatin was then de-crosslinked and purified by phenol:chloroform extraction. DNA concentration was measured using QuantiT PicoGreen (Life Technologies), and 40 μg of DNA was sheared to an average size of 400 bp, using the manufacturer's instructions (Covaris). The sheared DNA was end repaired, adenine tailed and double size-selected using AMPure XP beads to isolate DNA ranging from 250 to 550 bp. Ligation fragments marked by biotin were immobilised using MyOne Streptavidin C1 DynaBeads (Invitrogen) and ligated to paired-end adaptors (Illumina). The immobilised HiC libraries were amplified using PE PCR 1.0 and PE PCR 2.0 primers (Illumina) with 7 PCR amplification cycles. Capture HiC of promoters was carried out with SureSelect target enrichment, using the custom designed biotinylated RNA bait library and custom paired end blockers according to the manufacturer's instructions (Agilent Technologies). Biotinylated 120mer RNA baits were designed to the ends of HindIII restriction fragments overlapping Ensembl annotated promoters of protein coding, non-coding, antisense, snRNA, miRNA and snoRNA transcripts[36]. A target sequence was accepted if its GC content ranged between 25 and 65%, the sequence contained no more than two consecutive Ns and was within 330 bp of the HindIII restriction fragment terminus. A total of 22,076 HindIII fragments were captured, containing a total of 31,253 annotated promoters for 18,202 protein coding and 10,929 nonprotein genes according to Ensembl v75 (http://grch37.ensembl.org). After library enrichment, a post capture PCR amplification step was carried out using PE PCR 1.0 and PE PCR 2.0 primers with 4 PCR amplification cycles. PCHi-C libraries were sequenced on the Illumina HiSeq 2500 platform3 50-PE sequencing lanes per PCHi-C library. Sequencing reads were processed and mapped with HiCUP (version 0.5.5) and PCHi-C interactions were called using CHiCAGO (version 0.2.5) with default parameters[33].

**Genotyping check of ChIP-Seq and PCHi-C data.** Identity matching for each sample and for each analysis was performed by extracting genotypes from PCHi-C and ChIP-seq and comparing them to SNPs from the WGS data. The first stage of verifying the sample identity concordance between the ChIP-seq and WGS data involved pre-processing the BAM files for one autosomal chromosome (chr1) to remove PCR duplicates and reads with mapping quality score <10. The variants were then called from the resulting BAM file using *mpileup* from the SAMtools package[77]. The variants with QUAL < 20, DP < 5 and GQ < 5 were filtered out. Then, we compared genotypes of the filtered variants with genotypes generated from WGS and imputation. The genotypes generated were considered to be from the same sample if the concordance rate was greater than 90%.

**Allele-specific analysis of TF binding.** For allele-specific analysis, we used the phased WGS VCF that was also utilised for QTL mapping but here we removed indels and only considered biallelic single nucleotide variants. We then mapped deduplicated ChIP-seq reads on each allele of each SNVs using GATK ASEReadCounter with default parameters, base quality ≥2 and mapping quality ≥15. We then filtered for heterozygous SNVs only with ≥10 read counts per site and nonzero counts in both alleles. We required two donors meeting these read counts criteria at each site. To carry out association analysis, we used Rasqual[26] with total read counts per sample as an offset parameter. Note that Rasqual uses a model that corrects for reference mapping bias and genotyping errors. To correct for non-genetic confounders, we applied PCA with and without permutation on normalised read counts in log2RPM across all sites and picked the first $N$ components whose explained variances are greater than those from permutation as covariates for Rasqual. Finally, we only considered SNVs found within peaks to determine the direct allele-specific effect on TF binding of PU.1 and CTCF in neutrophils.

**Allele-specific analysis of PCHi-C.** The genotypes of PCHi-C donors were obtained from Cambridge Bioresource phase 4 (Illumina core exome chip). We phased the genotype using BEAGLE2 (v2.0.5)[78] and imputed using Positional Burrows–Wheeler Transform and Haplotype Reference Consortium (release 1.1) as reference panel, via the Sanger imputation service. We then filtered sites for ≥5% minor allele frequency, HWE $p$ value ≥ $1 \times 10^{-6}$, ≤5% sample missingness and INFO score ≥0.8. We removed indels and only considered biallelic single nucleotide variants. We used WASP[79] to remove PCHi-C reads that are likely to be biased towards the reference allele. We then mapped deduplicated PCHI-C reads on each allele of each SNVs using GATK ASEReadCounter with default parameters, base quality ≥2 and mapping quality ≥15. We then filtered for heterozygous SNVs only with ≥10 read counts per site and nonzero counts in both alleles. Finally, we only considered heterozygous sites with allele bias of ≤40% or ≥60%, after removing extreme bias of <1% or >100%.

**Enrichment analysis of tfQTLs and hQTLs in PIRs.** Each of these heterozygous SNVs was annotated based on whether they were located in a PIR and whether they were significant tfQTLs (PU.1 and CTCF; $p < 1 \times 10^{-5}$) or significant hQTLs (H3K27me3, H3K4me3, H3K27ac; $p < 1 \times 10^{-5}$). Fisher's exact tests were carried out separately for each sample and for each cell type to test for enrichment of tfQTLs and hQTLs that fall into PIRs. Finally, the mean and standard deviation were calculated across all samples for each cell type. In another approach, all samples were combined across both cell types. SNVs were removed if they were not observed in at least two samples, or in one sample and in the two cell types, or if the allelic ratio (REF reads/ALT reads) was not consistent across the samples or cell types. Enrichment was tested for SNVs where at least N samples fell into a PIR and at least N samples carried a significant tfQTL or hQTL for increasing number of samples $N$ ($N = 1, 2, 3, 4$).

**Enrichment of genome-wide association SNPs within ChIP-seq marked regions.** To test for significant enrichment of trait associated SNPs within regions of interest, we applied GWAS analysis of regulatory or functional information enrichment with LD (GARFIELD)[39]. H3K27ac and H3K4me1 occupied regions in neutrophils were obtained from[14]. Neutrophil annotations for PU.1, C/EBPβ, H3K4me3 and H3K27me3 were generated as described above. With the exception that H3K27me3, regions were not chunked into 2.5 kb bins. Monocyte annotations are obtained from data shown in Supplementary Fig. 6a for PU.1 and C/EBPβ.

**Colocalisation between diseases and molecular trait.** To overlap our QTL results to GWAS catalogue, we calculated the LD information based on our WGS data using plink v1.9[80]. For all the QTLs that either directly mapped to the GWAS variants or in LD ($r^2 \geq 0.8$), we considered that the QTL variant overlapped with a GWAS signal. For the cases where we further selected six autoimmune diseases, we took forward the overlapping disease variants with $p$ value ≤$5 \times 10^{-8}$ in six selected studies are coeliac disease[40], IBD[41], including CD and UC, multiple sclerosis[42], Type 1 diabetes[43], and RA[44]. The associations of IBD, CD and UC in the European cohorts were used for this study. We also used Type 2 diabetes[45] as a negative control. We used a Bayesian colocalization method[46,81] to elucidate whether the observed overlap between disease and molecular trait may be due to a shared genetic effect. The method calculates the PP, versus the null model of no association, for four alternative models: a model where a region or locus contains a single variant associated with either the molecular trait or disease (models 1 and 2); a model where a single variant affects association with both traits (model 3); or a model where two distinct associations exist (model 4). The method derives the PP of each variant in the locus being causal one under different models, and the PP of a given locus is then the integral sum of the PPs of all variants within, with all variants under equal prior probability to be causal. The prior for each model is computed to be one that maximises the log-likelihood function[46]. We acknowledge the limitations of the model: it assumes one causal variant in the locus; and in the case of high LD between two causal variants the model has limited power to distinguish model 4 from model 3. We also note that colocalisation does not imply a causal relationship between molecular trait and diseases, but may be compatible also with the same variant having independent ('pleiotropic') effects on molecular traits and disease. We applied a colocalisation test for each of the 1003 disease-molecular trait pairs, where the lead SNPs in both traits are in $r^2 \geq 0.8$. To avoid overlapping 2 Mb-wide genetic loci due to features in close proximity (e.g., splicing junctions, genes, histones peaks, CpGs in islands), we tested colocalisation per locus, which means that the prior model parameters were estimated using one locus instead of multiple loci and hence the priors may be overestimated.

**Reporting summary.** Further information on research design is available in the Nature Research Reporting Summary linked to this article.

## Data availability
Data generated in this study was deposited to the European Genome-phenome Archive under the following accession IDs: transcription factor data: EGAD00001004571; H3K4me3: EGAD00001002711; H3K27me3: EGAD00001002712; PCHi-C:

EGAS00001001911. Data from UK Biobank GWAS graciously provided by the Neale lab, [http://www.nealelab.is/uk-biobank] UKBB GWAS Imputed v3 - File Manifest Release 20180731. All additional data are available from authors upon request. Source data are provided with this paper.

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

## Acknowledgements

This work was supported by the Wellcome Trust grant reference (098051, 206194 and 108749/Z/15/Z), EU-FP7 Project BLUEPRINT (HEALTH-F5-2011-282510). This research was supported by the NIHR Cambridge Biomedical Research Centre (BRC-1215-20014). The views expressed are those of the author(s) and not necessarily those of the NIHR or the Department of Health and Social Care. M.S. is core-funded by the Medical Research Council of the UK as an MRC Investigator (MC-A652-5QA20). B.M.J. is funded by the FEDER/Spanish Ministry of Science and Innovation (RTI2018-094788-A-I00) and by La Caixa Banking Foundation Junior Leader project (LCF/BQ/PI19/11690001). D.R., A.D., L.C., and P. Flicek are supported by the European Molecular Biology Laboratory. We gratefully acknowledge the participation of all NIHR Cambridge BioResource volunteers. We thank the Cambridge BioResource staff for their help with volunteer recruitment. We would like to thank members of the EU Blueprint consortium including, Dirk Paul, Daniel Rico, Vera Pancaldi, Willem Ouwehand and Henk Stunnenberg. In addition, Stephen Wingett from Babraham Institute Bioinformatics team. Members of the sequencing centres of Wellcome Sanger Institute and Babraham Institute. Lorenz Wernisch for advice on statistics.

## Author contributions

Conceived and designed the study, S.W., B.M.J., M.S and N.S. Performed experiments, S.W. and B.M.J. Generated experimental resources, F.B., S.F. and B.F. Performed formal analysis, S.W., L.V., A.L.M., K.K., L.C., Y.Y., S.E., V.I., H.E., M.T., D.R., A.D. and M.S. Investigation, S.W., L.V., K.W. and A.L.M. Data curation, L.V., Y.Y., H.P., D.R., A.D. and L.C. Supervision and study coordination, P. Flicek., L.C., K.D., T.P., P. Fraser., M.F., M.S. and N.S. Project Administration, D.M., L.C., K.D., P.Fraser., M.F., M.S. and N.S. Performed primary manuscript writing S.W., A.L.M., M.S. and N.S.

## Competing interests

P. Fraser. and M.S. are co-founders of Enhanc3D Genomics Ltd. P. Flicek is a member of the scientific advisory board of Fabric Genomics, Inc., and Eagle Genomics, Ltd. The remaining authors declare no competing interests.
