## [Peer Review File · Nature Communications]

Reviewers' Comments:

Reviewer #2:

Remarks to the Author:

The manuscript by Watt et al. is a novel, appealing study on chromatin biology and the genetically determined variation in PU.1 recruitment to DNA in human neutrophils and its association with complex autoimmune disease traits. The study is of broad interest and provides valuable omics datasets in a "demanding to work with" cell type with a prominent role in human health and disease. The data are presented in an understandable way and the manuscript is well written.

Major concerns:

1. ChIP seq quality (Suppl. Table 2): TF and histone ChIP-Seq datasets are the core of the manuscript, however several samples on which ChIP-sequencing was done do not fulfill the ENCODE guidelines for TF and histone ChIP seq.

- Sequencing depth: there are many samples, e.g. 1/3 of all samples in PU.1 ChIP-seq that have extremely low (< 5 mio) or insufficient (5-10 mio) number of the usable reads, similarly for H3K4me3 only one sample achieves 20 mio usable reads (recommended depth for narrow histone peaks, if based on Q15 read numbers)

- Variation in sequencing depth is rather large between samples as well as between neutrophils and monocytes for the same TF

- Number of duplicates in some samples represents almost half of all reads, did the cell number vary significantly between samples (cell death?).

The authors should discuss these limitations and their potential effects on the presented results.

2. The authors should clearly substantiate the rationale behind studying the genetically determined variation in genomic binding of PU.1 in association with the selected panel of autoimmune diseases, e.g. the known roles of PU.1 in these autoimmune diseases, pure discovery study etc... There are few examples of given PU.1 QTL and their effects on gene expression in association with selected traits (Figs. 4d, 5b, Suppl. Fig. 6). Please provide additional evidence (if known from literature) how the specified gene expression changes could shape a particular trait. Given the extensive role of neutrophils in SLE it would be of high interest to provide a SLE GWAS example (as in Fig. 4d) and a circus plots for fold enrichment of SLE GWAS loci within functional annotations derived from TF binding/histone modifications in neutrophils and monocytes (as in Fig 5a.).

3. The title is an overstatement, the study is descriptive and does not provide direct experimental evidence that the variation in PU.1 binding and chromatin looping influences the autoimmune disease susceptibility. For this, further functional experiments would be necessary beyond mere gene expression changes, e.g. at least showing differential disease-relevant functional behaviour of the cells in different genotypes driven by the affected gene expression and PU.1 variation at given locus.

Minor comments:

- Lines 188-192, Fig. 4 a,b and Suppl. Fig. 4f: please comment on the heterozygous SNPs within PIRs with evidence of allelic bias enriched for other histone marks apart from H3K27me3

- Figs. 4d, 5b, Suppl. Fig. 6: please zoom into genomic SNP regions to provide a good insight into the distance relationships between different histone/TF peaks

- Fig.1a, Suppl. Table 2 – numbers of ChIP samples do not fit (e.g. for PU.1 in neutrophils): which samples were excluded and why? Not all ChIP samples reported in the Suppl. Table 2 meet the QC inclusion criteria (RSC>0.8 and NSC>1.05), were all reported samples kept in analysis? The gender of the same individual is sometimes unclear – reported both as male and female. Please explain the QC abbreviations e.g. Q15 reads etc.

- Fig. 1d: Based on which criteria a subset of detected PU.1 QTLs was selected for allele-specific association analysis?

- ChIP-seq: Report antibody amounts used for ChIP-seq. What genome assembly was used to align ChIP-seq datasets? Why two different antibodies against PU.1 were used? Why two different Illumina

platforms were used for ChIP-seq and how the samples were allocated to each of them?

- Suppl. Fig. 1 a-c shows QC passed aligned reads. Which QC does this apply to – Q15 reads?
- Suppl. Fig. 1b shows consistently more PU.1 and CEBPB binding sites in monocytes versus neutrophils, but the sequencing depth of the usable reads is smaller for neutrophils versus monocytes, authors should comment on this.
- Please check if all references are cited.

Reviewer #3:

Remarks to the Author:

Watt and colleagues performed ChIPseq of the myeloid transcription factor PU.1 in neutrophils from nearly one hundred people. They have evaluated the extent of which genomic SNP variants at PU.1 bound sites would affect chromatin, gene expression and, as a result of this, neutrophil immune functions and disease involvement.

This study is for its main part technically well performed and the statistical analyses, as far as I can tell, have been adequately performed. However, although interesting, many of the presented findings are not really surprising and are predictable from the current literature. An example of this is the well-known functional association of PU.1 with CEBPbeta to control myeloid genes.

Although based on a high sample size allowing strong associations, the manuscript is purely descriptive, describing statistic-based correlations. Biological experiments to validate the findings are missing. In my opinion, the impact of this manuscript would greatly increase if the authors test at least some of the identified PU.1-associated SNP variants with functional experiments to assess if a biological effect is measurable. They could do this with the target genes mentioned from line 215 onwards. The CPEB4 gene would, for example, be a good option. Using CRISPR/Cas9-based genome editing, the authors could address if the rs79137-C SNP variant does indeed affect PU.1 binding that much that expression of the CPEB4 gene is altered and that neutrophil proliferation is impeded. A current study in mice (Fischer et al. 2019) has shown that PU.1 represses at least as many genes in neutrophils than it stimulates, thereby impeding some neutrophil immune functions. This repressive role of PU.1 is not considered by the authors of this manuscript.

Line 178: Why did the authors use PChi-C data from Chen et al. and not their own data?

Line 191: An alternative interpretation could be that the chromatin architecture shapes the Polycomb-associated histone mark.

Reviewer #4:

Remarks to the Author:

This paper offers a wealth of data, and is a valuable resource to other groups as part of an extended 'blueprint' initiative. The analysis pipeline is now well established for this type of work, generating and overlaying different layers of QTL data, chromatin interaction data and GWAS data to demonstrate correlation and enrichment. As such, this well established pipeline is adhered to perfectly well, with adequate QC and statistical rigour.

The Capture Hi-C feels very 'off the shelf' and not tailored for this particular experiment. 50% of the tfQTL SNPs are 2.5kb-1Mb away from the PU.1 peaks. I think it would be more novel, and make a more interesting manuscript, if these were targeted by a capture system - for example do SNPs >100kb away from the PU.1 peak physically interact with the peak itself? What is the mechanism for this observed phenomena? You state on line 128/129 that there may be more "complex and cell-type specific long distance chromatin contacts". It may be that these long range interactions are not validating in a second cell type because of the low affect size of distal SNPs, low sample number and

consequent low power. Alternatively, are these SNPs genuinely interacting with the PU.1 peaks in a cell type specific manner? This seems one of the more interesting/novel aspects of this work, that this group has the capability to address.

Ref: NCOMMS-19-15090A

Watt et al - Reviewers' comments

Reviewer #2 (Remarks to the Author):

The manuscript by Watt et al. is a novel, appealing study on chromatin biology and the genetically determined variation in PU.1 recruitment to DNA in human neutrophils and its association with complex autoimmune disease traits. The study is of broad interest and provides valuable omics datasets in a “demanding to work with” cell type with a prominent role in human health and disease. The data are presented in an understandable way and the manuscript is well written.

We thank the Reviewer for the positive overall assessment of our manuscript and helpful suggestions, addressed below.

Major concerns:

1. ChIP seq quality (Suppl. Table 2): TF and histone ChIP-Seq datasets are the core of the manuscript, however several samples on which ChIP-sequencing was done do not fulfill the ENCODE guidelines for TF and histone ChIP seq.

- Sequencing depth: there are many samples, e.g. 1/3 of all samples in PU.1 ChIP-seq that have extremely low (< 5 mio) or insufficient (5-10 mio) number of the usable reads, similarly for H3K4me3 only one sample achieves 20 mio usable reads (recommended depth for narrow histone peaks, if based on Q15 read numbers)

- Variation in sequencing depth is rather large between samples as well as between neutrophils and monocytes for the same TF

The authors should discuss these limitations and their potential effects on the presented results.

Thank you for raising these important points, related to the challenge of generating a dataset of this scale. To provide more detail, data was collected in three main tranches over a 24 month period. First, we performed sequencing on a HiSeq2000 machine at a high plex level. However as data output was low in some cases (we aimed for a minimum 12 million total reads), we moved to HiSeq2500 at lower plex in order to achieve a minimum of 15 million total reads.

In response to the Reviewer's concerns, we have performed additional quality control checks which further support the validity of our QTL analyses despite differences in coverage. The ENCODE guidelines from Landt et al (Genome Research 2012) suggest calculating the Fraction of Reads in Peaks (FrIP) as a measure of ChIP enrichment. We have now included this metric as a **new Supplementary Figure 1c**. As expected, FrIP positively correlates with the number of peaks called. According to Landt et al, the ENCODE consortium “scrutinised” the 25% of data sets where FrIP falls below 1% (at the time of publication). Our median FrIP enrichment was 16% for PU.1, and only two samples had a score of 5%. We also plot the number of peaks called versus the total number QC passed reads; as expected, no correlation was observed (**Supplementary Figure 1c**).

It should be noted that the QTL analysis was only performed on the most robust set of ChIP enriched regions, following the strategy we previously employed in Chen et al. (Cell 2016). For inclusion in the reference peakset (phenotype matrix), we only considered peaks with log2RPM > 0 in at least

50% of samples. This stringent cut off had the effect of reducing the number of PU.1 binding sites passed for the association test from 99,000 to 36,000.

Our QTL phenotype matrix was quantile normalised and batch corrected using PEER (<https://github.com/PMBio/peer>). Differential binding analysis of PU.1 and C/EBP β between neutrophils and monocytes presented in **Supplementary Figure 6a** was performed out of this framework. These analyses were performed on the ChIP-seq datasets with the largest number of detected peaks and FrIP values for each factor and cell type. The DiffBind (<http://bioconductor.org/packages/release/bioc/vignettes/DiffBind/inst/doc/DiffBind.pdf>) package we used to perform the differential binding analysis provides quality control plots as part of the process which we now include as **new Supplementary Figures 5a and b**. These QC plots demonstrate the overall high agreement across individuals for the same cell types and TFs.

Genotypes are a fixed component and should be randomly distributed and not associated by batch. As is standard practice for QTL experiments, the phenotype matrix was quantile normalised and we applied PEER correction to account for batch effects. To alleviate the Reviewers' concerns we include new figures demonstrating that variance in the data acquisition has been accounted for, and does not hamper our results or interpretations. PU.1 data was generated in three batches and these were sequenced at a median depth of 12.56, 19.97 and 51.14 million total reads, respectively. We compared the QTL effect sizes detected on an interim analysis with 67 donors from the first two batches with the full analysis and observed strong agreement between the two (**new Supplementary Figure 2c**). As effect sizes (betas) could be compromised by limited dynamic range or signal to noise ratio, this analysis demonstrates that adding samples with a large variance in read depth does not skew the beta. As expected, adding more donors increased the power to detect QTLs, yielding an additional 642 PU.1 QTLs passing the significance threshold (**new Supplementary Figure 2d**). As an illustrative example we include a **new Supplementary Figure 2e** showing normalised PU.1 signal at the rs791357 QTL, demonstrating that the observed genetic effect is not confounded by any particular batch.

We also performed an additional validation of our PU.1 tfQTLs using allelic imbalance detection with RASQUAL (Kumasaka, Nat Genet 2016). This analysis has validated QTLs 98.8% and 95.5% of peaks associated with proximal and distal variants, respectively (**Figure 1d**).

- Number of duplicates in some samples represents almost half of all reads, did the cell number vary significantly between samples (cell death?).

Cells were formaldehyde cross-linked at the same time as those collected for ChIP-seq and RNA-seq published in Chen et al. (Cell 2016). RNA-seq analysis did not suggest that there was any batch effect driven by cell death. Neither did we observe specific samples consistently generating high duplicate read numbers across antibodies. The high number of duplicate reads in some ChIP-seq experiments is likely due to stochastic variation in pull-down efficiency coupled to standardised PCR conditions during library preparations, rather than a systematic deficiency in some donor samples. However, it is important to note that duplicate reads were collapsed prior to constructing the phenotype matrix.

2. The authors should clearly substantiate the rationale behind studying the genetically determined variation in genomic binding of PU.1 in association with the selected panel of autoimmune diseases, e.g. the known roles of PU.1 in these autoimmune diseases, pure discovery study etc... There are few examples of given PU.1 QTL and their effects on gene expression in association with selected traits (Figs. 4d, 5b, Suppl. Fig. 6). Please provide additional evidence (if known from literature) how the specified gene expression changes could shape a particular trait. Given the extensive role of neutrophils in SLE it would be of high interest to provide a SLE GWAS example (as in Fig. 4d) and a circus plots for fold enrichment of SLE GWAS loci within functional annotations derived from TF binding/histone modifications in neutrophils and monocytes (as in Fig 5a.).

We thank the Reviewer for this suggestion. We have now expanded the Introduction and included references allowing us to set out key concepts. In addition, we have restructured the manuscript placing more emphasis on trait associations. We also include new analysis investigating GWAS generated from UK Biobank phenotypes by the Neale laboratory (**new-Supplementary Figure 9**). This analysis reveals an enrichment of PU.1 tfQTLs at traits such as asthma and hayfever, but, somewhat surprisingly, not at those associated with infections such *Clostridium difficile*. We also agree that we should have emphasised SLE more given the known relevance of neutrophils for this condition. We identify two colocalisations at the *IER3* locus on chromosome 6 (rs9262174 and rs2233956), however this region has a complex LD structure.

3. The title is an overstatement, the study is descriptive and does not provide direct experimental evidence that the variation in PU.1 binding and chromatin looping influences the autoimmune disease susceptibility. For this, further functional experiments would be necessary beyond mere gene expression changes, e.g. at least showing differential disease-relevant functional behaviour of the cells in different genotypes driven by the affected gene expression and PU.1 variation at given locus.

We agree that we did not directly demonstrate causality and have amended the title to “Genetic perturbation of PU.1 binding and chromatin looping at neutrophil enhancers associates with autoimmune disease susceptibility”.

Minor comments:

- Lines 188-192, Fig. 4 a,b and Suppl. Fig. 4f: please comment on the heterozygous SNPs within PIRs with evidence of allelic bias enriched for other histone marks apart from H3K27met3

Enrichment of hQTLs associated with allelic bias in PIRs was presented in **Figures 4b & c**, however we have now restructured the manuscript and these can now be found as **Supplementary Figure 7c-e**.

- Figs. 4d, 5b, Suppl. Fig. 6: please zoom into genomic SNP regions to provide a good insight into the distance relationships between different histone/TF peaks

We thank the reviewer for the suggestion and we have added zoomed-in windows around the lead SNP at corrected panels 5c, Supplementary 10a and 10b, as requested, permitting better insight into the relationship between the lead SNP and the TF-bound region detected by ChIP.

- Fig.1a, Suppl. Table 2 – numbers of ChIP samples do not fit (e.g. for PU.1 in neutrophils): which samples were excluded and why?

We are sorry for the misunderstanding, however we believe the numbers to be correct. One source of possible misunderstanding comes from the fact that the ChIP experiments that failed QC are not included in **Supplementary Table S1**. There are 93 PU.1 neutrophil individuals included in the association analysis as stated in **Supplementary Table S1**. There are 4 technical replicates which are included as part of the data release for PU.1 in neutrophils. Three of these were used in analysis included in **Figure S2a**. To make this more clear, we have added a sentence to the legend of **Figure S2a** that states: “For three individuals, PU.1 profiling was carried out in duplicate as independent technical replicates.” These experiments can be identified from column 4 in **Supplementary Table S1**.

Not all ChIP samples reported in the Suppl. Table 2 meet the QC inclusion criteria (RSC>0.8 and NSC>1.05), were all reported samples kept in analysis?

Inclusion criteria was as described in Chen et al. (Cell 2016) and all samples reported in **Table S1** were included in the QTL analysis. We removed ChIP samples that had a relative strand correlation (RSC) < 0.8 and normalised strand correlation (NSC) < 1.05. We defined high confidence datasets as those with RSC > 0.8 and NSC > 1.05. Otherwise, we used genome browser tracks to visually confirm a good ChIP and include it in the final data set.” Only 3 of our CTCF tfQTL datasets fell beneath this criteria and required manual inspection. For the Reviewer’s assessment we provide a UCSC genome browser session with the tracks listed below containing indicative examples of what we considered high-quality and sub-optimal, but acceptable data (http://genome.ucsc.edu/s/steve9110/NatComms_sbw). We now also report the fraction of reads in peaks for each dataset in **Table S1** and **new Supplementary Figure 1c**.

SampleID	Individual	Factor	Antibody	Treatment	Gender	Number #	TOTAL_READS	Q1_READS	Q1_DUPLICATES	Q15 no dup	NSC	RSC	FIP	Sequencing platform
NS1623	S011JH	CTCF	c1541021C	CD14-positive.CD16-negative.classical.monocy	Female	16941	61,994,046	52,144,625	687,810	50,648,644	1.049481	1.461645	0.06	HiSeq2500
NS1624	S011ER	CTCF	c1541021C	CD14-positive.CD16-negative.classical.monocy	Female	14433	33,463,657	27,806,521	632,568	26,727,699	1.031826	1.076439	0.04	HiSeq2500
NS1610	S00XK6	CTCF	c1541021C	Mature Neutrophil	Female	9403	24,055,398	19,375,194	320,483	18,731,962	1.05272	0.721982	0.05	HiSeq2500
NS1682	S00PDF	CTCF	c1541021C	Mature Neutrophil	Female	46692	57,360,537	51,325,373	3,532,362	47,124,080	1.86885	1.588993	0.33	HiSeq2500
NS1522	S00YEE	PU.1	sc352	CD14-positive.CD16-negative.classical.monocy	Male	66665	25,460,999	22,201,170	475,567	21,426,554	1.5476	2.1161	0.11	HiSeq2500
NS1511	S00PDF	PU.1	sc352	CD14-positive.CD16-negative.classical.monocy	Female	92004	28,113,473	24,981,543	1,399,480	23,255,140	2.728	2.6007	0.21	HiSeq2500
NS1607	S011ER	PU.1	sc352	Mature Neutrophil	Female	11837	28,569,507	19,635,998	369,346	18,930,033	1.068669	0.867557	0.05	HiSeq2500
NS1681	S00PDF	PU.1	sc352	Mature Neutrophil	Female	61632	54,857,060	48,200,741	2,066,637	45,467,340	1.635255	2.416991	0.22	HiSeq2500
NS1558	S00T14	CEBPB	sc150	CD14-positive.CD16-negative.classical.monocy	Male	49005	18,681,839	16,322,570	479,774	15,586,185	1.9866	2.2267	0.11	HiSeq2500
NS1565	S00I0M	CEBPB	sc150	CD14-positive.CD16-negative.classical.monocy	Female	36887	13,477,478	11,538,613	215,664	11,140,784	1.614	2.0289	0.08	HiSeq2500
NS753	S00PJ	CEBPB	sc150	Mature Neutrophil	Female	15788	11,796,660	10,162,821	3,223,163	6,813,410	1.5788	3.2568	0.1	HiSeq2000
NS793	S00QWA	CEBPB	sc150	Mature Neutrophil	Female	17521	13,444,062	11,717,609	2,590,163	8,969,518	1.6082	3.2372	0.12	HiSeq2000

The gender of the same individual is sometimes unclear – reported both as male and female.

We thank the Reviewer for spotting this error - now corrected.

Please explain the QC abbreviations e.g. Q15 reads etc.

Q15 refers to the minimum mapping quality score (MAPQ), to make this more clear we have amended the methods : “Reads with mapping quality less than 15 (MAPQ 15) were removed (SAMtools v0.1.18).” In addition we have amended the headers in **Supplementary Table 1**.

- *Fig. 1d: Based on which criteria a subset of detected PU.1 QTLs was selected for allele-specific association analysis?*

For allele specific analysis, we used the phased WGS VCF that was also utilised for QTL mapping, but here we removed indels and only considered biallelic single nucleotide variants (please see Methods section). We then mapped deduplicated ChIP-seq reads on each allele of each SNVs using GATK ASEReadCounter with default parameters, base quality ≥ 2 and mapping quality ≥ 15 . We then filtered for heterozygous SNVs only with ≥ 10 read counts per site and nonzero counts in both alleles. We required 2 donors meeting these read counts criteria at each site.

- *ChIP-seq: Report antibody amounts used for ChIP-seq.*

To clarify, we used 2.5 μ g of antibody for all ChIP-seq reactions, (please, see Methods section).

What genome assembly was used to align ChIP-seq datasets?

We apologise for this omission and now mention in Methods that GRCh37 (hg19) was used.

Why two different antibodies against PU.1 were used?

We initially used sc-22805, however later in the project sc-352 was shown to give better ChIP enrichment in primary neutrophils. As the data for sc-22805 had passed QC, and batch correction was performed prior to the association test, it was deemed acceptable not to remove the data for these individuals. The 13 ChIPs using sc-22805 were among the 24 samples sequenced on the HiSeq2000 (see comparison with other batches in **new Supplementary Figure 2e**).

Why two different Illumina platforms were used for ChIP-seq and how the samples were allocated to each of them?

We moved to HiSeq2500 in order to gain more sequencing depth as the project progressed. We have added a column in **Supplementary Table 1** stating which samples were sequenced on each platform.

- *Suppl. Fig. 1 a-c shows QC passed aligned reads. Which QC does this apply to – Q15 reads?*

We apologise for not adequately describing our ChIP-seq QC figures. **Supplementary Figure 1a** applied to total mapped reads ($q \geq 0$) and we have now clarified this in the legend. The numbers of QC-passed reads (MAPQ ≥ 15) are now included in **new Supplementary Figure 1c**, showing passed reads versus FRiP and total peaks called.

- *Suppl. Fig. 1b shows consistently more PU.1 and CEBPB binding sites in monocytes versus neutrophils, but the sequencing depth of the usable reads is smaller for neutrophils versus monocytes, authors should comment on this.*

We thank the reviewer for their feedback. There could be a technical explanation for this observation, as indeed, as the Reviewer notes, neutrophils are a tricky cell type to handle. However, monocytes are

uncommitted cells and there is some evidence that they have more active epigenome and transcriptome compared with neutrophils (Rico et al; bioRxiv 2017, <https://www.biorxiv.org/content/10.1101/237784v1>). Consistent with this, the FrIP analysis (**new Supplementary Figure 1c**) shows that for both PU.1 and C/EBP β , monocyte data sets accrue a greater number of peaks than neutrophils for similar FrIP values. This suggests that monocytes may genuinely have more binding sites than neutrophils for the same factors.

- Please check if all references are cited.

We have extensively rewritten the Introduction and Discussion adding more references.

Reviewer #3 (Remarks to the Author):

Watt and colleagues performed ChIPseq of the myeloid transcription factor PU.1 in neutrophils from nearly one hundred people. They have evaluated the extent of which genomic SNP variants at PU.1 bound sites would affect chromatin, gene expression and, as a result of this, neutrophil immune functions and disease involvement.

This study is for its main part technically well performed and the statistical analyses, as far as I can tell, have been adequately performed. However, although interesting, many of the presented findings are not really surprising and are predictable from the current literature. An example of this is the well-known functional association of PU.1 with CEBPbeta to control myeloid genes.

Although based on a high sample size allowing strong associations, the manuscript is purely descriptive, describing statistic-based correlations. Biological experiments to validate the findings are missing.

We thank the Reviewer for appreciating the technical quality of our work. It is true that our study can be considered descriptive purely because it does not artificially perturb the system - which is challenging in primary cells. However, instead we leverage a huge amount of natural variation in these cells, essentially using it as “nature’s mutagenesis experiment” at a scale far exceeding what is currently possible with artificial perturbation. Specifically, the Blueprint EpiVar cohort of 200 individuals contains ~8 million common genetic variants, and from these we identify 12,000 top associated SNPs implicated in altered binding affinity of 2,031 PU.1 binding sites. Analysis of these data and their integration with additional genomic assays reveals mechanistic interpretation and transcriptional process at hundreds of loci. While this approach does have its own challenges and limitations, it enables us to sample a very large number of perturbations globally in an unbiased manner, obtaining a genome-wide view of how genetically-determined alterations in TF binding underpin gene expression and disease susceptibility.

In my opinion, the impact of this manuscript would greatly increase if the authors test at least some of the identified PU.1-associated SNP variants with functional experiments to assess if a biological effect is measurable. They could do this with the target genes mentioned from line 215 onwards. The CPEB4 gene would, for example, be a good option. Using CRISPR/Cas9-based genome editing, the authors could address if the rs79137-C SNP variant does indeed affect PU.1 binding that

much that expression of the CPEB4 gene is altered and that neutrophil proliferation is impeded.

We agree that it would in principle be an interesting experiment that could illustrate the power of our approach on a small number of selected examples. However, it is a highly time- and resource-consuming task in our setting, particularly in primary cells, and one that has been infeasible due to the COVID situation, with our labs closed for a prolonged period. More generally, the added value of candidate-based perturbation experiments to support the validity of high-throughput datasets requires careful consideration (Hughes, J Biol 2009, doi: 10.1186/jbiol104). Influenced by the reviewers suggestion to investigate the results reported in Fischer et al (see below) that PU.1 is involved in dampening an overactive immune response to infection, we expanded our investigation beyond published autoimmune traits to include recently released GWAS summary statistics from the Neale lab using UK Biobank phenotypes. These phenotypes included self reported susceptibility to infections or the use of antibiotics and additional autoimmune traits such as allergies and rheumatoid arthritis. We observed no enrichment of our tfQTL SNPs within traits associated with infection or antibiotic use, however for the autoimmune traits we included we did observe inflation of our associated *p* values at these SNPs. These results have now been included in **new Supplementary figure 9**.

A current study in mice (Fischer et al. 2019) has shown that PU.1 represses at least as many genes in neutrophils than it stimulates, thereby impeding some neutrophil immune functions. This repressive role of PU.1 is not considered by the authors of this manuscript.

This is an excellent suggestion and we thank the reviewer for bringing the Fischer *et al* study to our attention. Initially we did not investigate instances where PU.1 tfQTLs and H3K27me3 hQTLs were positively correlated. We have now included a **new Figure 3a** plotting the correlation of effect sizes between PU.1 tfQTL our 2 hQTL datasets. The correlation between the betas for PU.1 tfQTL and repressive H3K27me3 was $r=-0.45$, suggesting PU.1 tfQTLs and H3K27me3 QTLs act in the opposite direction, however there are 131 QTLs that are positively correlated. In addition, for shared tfQTL and eQTL 128/905 eGenes are negatively correlated with PU.1 binding. We have added these results to our manuscript.

One example of a gene with a shared PU.1 tfQTL and eQTL showing opposite direction of effect is *RNASET2*. This is particularly interesting as the tfQTL/eQTL SNP in question (rs2149092) is also a known causal GWAS SNP associated with IBD and Crohn's disease and affects the canonical PU.1 motif (Gonsky *et al*; Gastroenterology 2017). We now include this example as **new Supplementary Figure 8**.

We planned to explore the functional effects of differential PU.1 binding at this locus in light of the findings of Fischer et al. This study proposes that the repressive effect of PU.1 binding is achieved by its ability to block the binding of transcriptional activator c-Jun upon immune response. Our plan was to initially knock down PU.1 in neutrophil cell line HL60, stimulate these cells with zymosan and study the effect of this perturbation on (a) c-Jun binding at the rs2149092-associated site (b) *RNASET2* expression. If these experiments supported our hypothesis, we would consider following this up further in a more targeted way, such as 'flip' the rs2149092 allele by CRISPR editing in

HL60 instead and/or recall blood donors by genotype and validate the effects in activated primary cells.

Unfortunately, our experiments were interrupted at an early stage by the closure of our labs due to the COVID-19 situation. Given the uncertainty of when the labs will return to normal operating conditions, and with the estimations suggesting an extension of the lockdown for many months, we have taken the difficult decision to take these analyses out of scope of this paper.

Line 178: Why did the authors use PCHi-C data from Chen et al. and not their own data?

We are sorry to not be clear enough in the text. The PCHi-C data did not in fact come from Chen et al., it was a combination of those published in Javierre et al and those generated for the present study. Line 178 refers to the gene expression QTL set that was indeed produced from the same cohort of donors as that published in Chen et al., 2016.

Line 191: An alternative interpretation could be that the chromatin architecture shapes the Polycomb-associated histone mark.

We agree that the causality of this effect cannot be fully established from our observation. However, there is a bulk of previous evidence across species demonstrating the role of Polycomb in establishing 3D chromosomal contacts, which is what we have alluded to in the text.

Reviewer #4 (Remarks to the Author):

This paper offers a wealth of data, and is a valuable resource to other groups as part of an extended 'blueprint' initiative. The analysis pipeline is now well established for this type of work, generating and overlaying different layers of QTL data, chromatin interaction data and GWAS data to demonstrate correlation and enrichment. As such, this well established pipeline is adhered to perfectly well, with adequate QC and statistical rigour. The Capture Hi-C feels very 'off the shelf' and not tailored for this particular experiment.

We thank the Reviewer for appreciating the value of our dataset and technical quality of our analyses. We agree that following our pilot work (Javierre et al., Cell 2016), Promoter Capture Hi-C is becoming a method of choice for linking GWAS variants to their target genes. In this study, we aimed to use this technique for the same purpose, although we agree that our multi-individual ChIP data prompts other questions such as those highlighted by the Reviewer below, for which this technique may not be optimal.

50% of the tfQTL SNPs are 2.5kb-1Mb away from the PU.1 peaks. I think it would be more novel, and make a more interesting manuscript, if these were targeted by a capture system - for example do SNPs >100kb away from the PU.1 peak physically interact with the peak itself? What is the mechanism for this observed phenomena?

We agree that this is a very exciting question. To assess the potential of a targeted PU.1 capture experiment to gain insight into this problem, we focused on PU.1 peaks and tfQTL SNPs located at gene promoters as this subset of loci are amenable to the analysis suggested by the Reviewer with the existing Promoter Capture Hi-C design. As shown in **Reviewer Fig 1**

below, we found that only a small minority of such regions engage in physical connections between a tfQTL SNP and a cognate distal PU.1 peak. While this analysis based on our existing data can only consider the special case, whereby at least one of the PU.1 peak or its cognate tfQTL localise to a gene promoter, it is in our view rather unlikely that PU.1 peak-tfQTL pairs in other locations would show a higher incidence of direct looping. For this reason, we felt that the time and resources required to design and optimise a targeted PU.1 Capture Hi-C experiment would not justify the potential benefits of this analysis.

In addition we found that PU.1 tfQTL SNPs whose cognate PU.1 binding sites localise to gene promoters show elevated eQTL effect sizes for the corresponding genes compared with the tfQTL SNPs whose cognate PU.1 binding sites are not promoter-based (**Reviewer Fig 2 below**). This suggests that PU.1 peak/tfQTL pairs associated with promoter-enhancer relationships have the strongest gene regulatory effects, compared with potential enhancer-enhancer interactions that are undetectable with our current promoter capture design.

Reviewer Figure 1. Proportion of direct contacts between PU.1 tfQTLs and their cognate PU.1 binding peaks for the ascertainable scenarios where either the tfQTL or peak lies in a baited promoter region. The results are shown for monocytes (left two bars; shades of green) and neutrophils (right two bars; shades of blue). For each cell type, the left-hand bars show how many distal (>25kb) PU.1 binding peaks overlap a baited promoter region (bait) and in how many of these cases any of the associated tfQTLs lies in the corresponding PIR as detected by PCHiC. The right-hand bars show in how many cases any distal tfQTL lies in a baited

promoter region (bait) and how often the associated PU.1 binding peak overlaps the corresponding PIR as detected by PCHiC.

Reviewer Figure 2. The effect sizes of eQTLs overlapping with PU.1 tfQTLs depending on the location of the cognate PU.1 binding peak. Density plots of gene expression QTL absolute effect size (beta) for PU.1 tfQTLs/eQTLs ($p < 1e-5$ for both) that lie within PIRs (dark cyan) detected in monocytes (broken lines) or neutrophils (dotted lines), as well as for the subsets of these QTLs corresponding to following three scenarios. Light blue: The associated PU.1 peak does not overlap any of the baited promoter regions. Sea green: the associated PU.1 peak overlaps another baited promoter region that does not show significant interactions with the corresponding PIR according to PCHiC. Dark blue: the associated peak overlaps the baited promoter that has a significant interaction with the corresponding PIRs according to PCHiC.

To gain insight into the factors underpinning tfQTL effects we generated CATO (Contextual Analysis of TF Occupancy; Maurano et al., Nat Genet 2015) scores for the subset of these loci that colocalised with GWAS signals (**Supplementary Figure 11**). CTCF motifs showed an enrichment in the CATO analysis. To investigate whether altered CTCF binding could explain some of our distal PU.1 tfQTLs we asked for how many shared associations was the lead SNP proximal to a CTCF binding site. We identified only three SNPs (rs11113432, rs72794132 and rs4746822 shown in **Reviewer Fig 3** below) where the lead SNP was proximal to a CTCF site. However, the low number of these shared effects could be due to the limited power of our CTCF QTL as a discovery set.

Reviewer Figure 3. Genome browser views of the three loci where the lead PU.1 QTL SNP is proximal to a shared CTCF tfQTL. Vertical dashes (the track immediately below the coordinate scale): locations of lead proxy SNPs. Dark red: the location of the CTCF site. Navy: the location of the shared PU.1 tfQTL (also highlighted by arrows). Green track marks the location of neutrophil H3K27ac hQTLs.

You state on line 128/129 that there may be more "complex and cell-type specific long distance chromatin contacts". It may be that these long range interactions are not validating in a second cell type because of the low affect size of distal SNPs, low sample number and consequent low power.

This analysis only includes the five matched donor samples where we had PU.1 ChIP-seq data in both neutrophils and monocytes. It would not be expected that power or sample size would affect the result. We agree, however, that the statement "complex and cell-type specific long distance chromatin contacts" is potentially misleading, and have rephrased this statement in the main text to: "This data suggest that proximal tfQTL variants in related cell types operate under a similar cis regulatory mechanism, whereas long range allelic control decays with distance and distal QTLs exhibit increased cell type specificity (Wong, *Nat Comms* 2017; Delaneau, *Science* 2019)".

Alternatively, are these SNPs genuinely interacting with the PU.1 peaks in a cell type specific manner?

The comment by the reviewer relating to the result presented in **Figure 2d**, is an interesting one and it is difficult to elucidate a mechanism for long-range allelic control here. To minimise the effect of cell type specific binding influencing this observation we included only shared PU.1 and C/EBP β tfQTL

(33 distal and 43 proximal) that had evidence of binding in both cell types, determined by a 1bp peak intersection with the monocyte dataset. These sites are a subset of those shown in (**Supplementary Figure 4a**).

This seems one of the more interesting/novel aspects of this work, that this group has the capability to address.

Reviewers' Comments:

Reviewer #2:

Remarks to the Author:

The authors addressed my concerns in detail. I have no further comments.

Reviewer #3:

Remarks to the Author:

I thank the authors for addressing my points of concern. Although I still think that adding functional data would have improved the overall significance of this study, I understand that laborious experiments impose a significant challenge during this pandemic. That being said, I think that this paper is strong enough even without further experimental validation, and will as it is for sure provide a significant contribution to the field.

Reviewer #4:

None